



# Quantifying Variability in Lagrangian Particle Dispersal in Ocean Ensemble Simulations: an Information Theory Approach

Claudio M. Pierard [1], Siren Rühs [1], Laura Gómez-Navarro [1,2], Michael Charles Denes [1], Florian Meirer [3], Thierry Penduff [4], and Erik van Sebille [1]

[1]Utrecht University, Institute for Marine and Atmospheric Research, Princetonplein 5, 3584 CC Utrecht, Netherlands
[2]Mediterranean Institute of Advanced Studies (IMEDEA, UIB-CSIC), Esporles, Spain
[3]Utrecht University, Debye Institute for Nanomaterials Science & Institute for Sustainable and Circular Chemistry, Inorganic Chemistry and Catalysis, Universiteitsweg 99, 3584 CG Utrecht, Netherlands
[4]Université Grenoble Alpes, CNRS, INRAE, IRD, Grenoble INP, Institut des Géosciences de l'Environnement (IGE), Grenoble, France

**Correspondence:** Erik van Sebille (e.vansebille@uu.nl)

**Abstract.** Ensemble Lagrangian simulations aim to capture the full range of possible outcomes for particle dispersal. However, single-member Lagrangian simulations are most commonly available and only provide a subset of the possible particle dispersal outcomes. This study explores how to generate the variability inherent in Lagrangian ensemble simulations by creating variability in a single-member simulation. To obtain a reference for comparison, we performed ensemble lagrangian simulations by advecting the particles from the surface of the Gulf Stream, around $35.61°$N, $73.61°$W, in each member of theensemble to obtain trajectories capturing the full ensemble variability. Subsequently, we performed single-member simulations with spatially and temporally varying release strategies to generate comparable trajectory variability and dispersal. We studied how these strategies affected the number of surface particles connecting the Gulf Stream with the eastern side of the subtropical gyre.

We used an information theory approach to define and compare the variability in the ensemble with the single-member strategies. We defined the variability as the marginal entropy or average information content of the probability distributions of the position of the particles. We calculated the relative entropy to quantify the uncertainty of representing the full-ensemble variability with single-member simulations. We found that release periods of 12 to 20 weeks most effectively captured the full ensemble variability, while spatial releases with a $2.0°$ radius resulted in the closest match at timescales shorter than 10 days. Our findings provide insights to improve the representation of variability in particle trajectories and define a framework for uncertainty quantification in Lagrangian ocean analysis.

## 1 Introduction

The ocean's dynamics, driven by atmospheric fluxes of energy and momentum at the surface, are characterized by phenomena that mutually interact across different spatiotemporal scales, including eddies, internal waves, zonal jets, and mixing processes, up to decadal and basin-scale fluctuations (Vallis, 2017). These multi-scale interactions are non-linear and difficult to model, presenting a significant source of uncertainty in Ocean General Circulation Models (OGCMs) and our understanding of ocean





circulation. Even under constant atmospheric forcing conditions, ocean models can produce divergent states from minimally

perturbed initial conditions (Penduff et al., 2014). This intrinsic variability becomes particularly prominent in eddy-permitting
models where small initial differences can cascade towards multi-decadal and basin scales (Grégorio et al., 2015; Leroux et al.,
2018; Zhao et al., 2023). To address these inherent uncertainties in OGCMs, researchers have increasingly adopted probabilistic
ensemble models, running multiple simulations with small perturbations to initial conditions or parameter values to capture
a broad range of possible ocean states (Penduff et al., 2018; Zanna et al., 2019). The ultimate goal of ensemble models is to
predict the probability density of the system's state at a future time (Leutbecher and Palmer, 2008).

Lagrangian particle tracking provides a powerful tool for studying ocean transport, mixing, and connectivity, with applica-
tions ranging from search and rescue operations (Breivik et al., 2013) to climate and environmental research (Bower et al.,
2019; Van Sebille et al., 2018). In these simulations, virtual particles are typically advected by velocity fields derived from
OGCMs, with their dispersal patterns intimately linked to the underlying ocean state. These advected particle trajectories are
chaotic, in which small perturbations in initial conditions or noise along their trajectories can lead to significant divergences
in particle trajectories (Koshel and Prants, 2006). The sensitivity to initial conditions is often used to generate variability in
particle trajectories to predict the drift of the particles when there is uncertainty in their initial conditions (Breivik et al., 2013).
An alternative approach to generating variability in the trajectories is to advect particles using a full ensemble of vector fields
or ensemble models, an approach followed from Melsom et al. (2012), in which they advected particles using an ensemble
of 100 members from the TOPAZ forecasting system. They found that ensemble average trajectories, calculated as the center
of gravity (mean position) of all ensemble members at each time step, are generally closer (on a straight line distance) to the
observed drifter trajectories than that from a deterministic single-member simulation. However, the study did not compare how
small perturbations in initial conditions in the single-member simulation performed relative to the trajectories advected by the
ensemble.

While ensemble Lagrangian simulations can capture a more complete spectrum of possible outcomes, single-member sim-
ulations, which sample only a subset of the possible outcomes, remain more prevalent due to computational constraints. In
operational oceanography, data assimilative models are commonly used to improve trajectory predictions by combining ob-
servations with model dynamics to find an optimal solution (Castellari et al., 2001). However, while assimilation can reduce
systematic biases and improve the mean state representation, it may not fully capture the underlying uncertainty and variability
in particle trajectories, particularly in regions with sparse observations (Jacobs et al., 2018). Our study addresses these limi-
tations by exploring ways of generating ensemble-like variability within single-member simulations. We assess performance
based on a connectivity analysis and dispersion patterns using a novel information theory approach. Our approach consists
of quantifying the variability in trajectories through the marginal entropy of particle position distributions and evaluating the
uncertainty in representing full-ensemble variability with single-member simulations.

We focused on the region east of Cape Hatteras in the North Atlantic Ocean, implementing spatially and temporally varying
release strategies to generate variability comparable to that observed in full ensemble simulations. This region was chosen to
study the connectivity of water parcels at the surface of the Gulf Stream with the Eastern North Atlantic and the subtropical
gyre. It was previously thought that the salty and warm surface water of the Gulf Stream feeds directly to the subpolar gyre.



However, recent Lagrangian studies have shown that the water parcels originating at the surface of the Gulf Stream recirculate within the subtropical gyre, becoming part of the subtropical mode water, and enter the subpolar gyre via sub-surface connections (Rypina et al., 2011; Burkholder and Lozier, 2014; Foukal and Lozier, 2016; Berglund et al., 2022). Our study builds upon these findings by quantifying how intrinsic ocean variability affects this connectivity pattern within the subtropical gyre, providing insights into the robustness and variability of these recirculation pathways.

## 2 Methodology

### 2.1 Model Set-Up

Lagrangian particles were advected offline using six years (2010-2015) of daily surface velocity fields produced by the North Atlantic NATL025-CJMCYC3 50-member ensemble simulation. This regional ensemble simulation was performed in the context of the OceaniC Chaos – ImPacts, strUcture, predicTability project (OCCIPUT), described in Penduff et al. (2014) and Bessières et al. (2017). This ensemble was performed using the NEMO v3.5 ocean/sea-ice model over the North Atlantic between 20°S and 81°N, with an eddy-permitting resolution of $1/4°$ and 46 vertical levels. The 50 ensemble members were initialized by the final state of a 15-year one-member spin-up that ended in December 1992. The inter-member dispersion was generated by activating a small stochastic perturbation in the equation of state during 1993 and deactivating it for the remaining simulation time. All ensemble members were driven by the same atmospheric forcing between 1993 and 2015, derived from the DRAKKAR Forcing Set 5.2 (DFS5.2; see Dussin et al. (2016)). The NATL025-CJMCYC3 1993-2025 simulation used here is similar to the NATL025-GSL301 1993-2012 simulation presented in Narinc et al. (2024), with one difference: tropical cyclones were enhanced in the forcing of NATL025-CJMCYC3 since they were too weak in DFS5.2. More details about the model setup are provided in Narinc et al. (2024).

These velocity fields were used to advect particles, where particle trajectories in each ensemble member were integrated using the Parcels framework v.3.0.2 (Delandmeter and van Sebille, 2019). Trajectories were integrated in three dimensions using a fourth-order Runge-Kutta scheme with a time step of 1 hour, storing the output with a daily timestep. We modeled passive particles (that is, particles that instantly adjust their velocity to that of the ambient flow) by only considering three-dimensional advection and ignoring all buoyant or diffusive forces. Additionally, particles that escaped the domain through the surface were placed back to a depth of $1\,\mathrm{m}$. We chose the region off the coast of Cape Hatteras as a study location because it is an important region where the Gulf Stream separates from the continental shelf and becomes a free jet (Mao et al., 2023; Buckley and Marshall, 2016).

This study explores methods to recreate the trajectory variability typically obtained from ensemble ocean simulations using only a single ensemble member. Figure 1 illustrates both the challenge and our proposed approaches. When particles are released from a fixed point (35.61°N, 73.61°W; yellow square) and tracked using different ensemble members, their trajectories (shown in black) diverge due to variations in the velocity fields. Our goal is to reproduce this dispersion using just one ensemble member.





We tested two approaches to achieve ensemble-like variability with single-member simulations by leveraging the sensitivity
to initial conditions. The first strategy varies the release locations spatially (shown in purple in Figure 1), creating a cloud of
initial positions centered around $35.61°$N and $73.61°$W. The purple circles indicate the varying release locations, while the
purple arrows show their subsequent trajectories. The second strategy (shown in orange) maintains the fixed release location
(yellow square) but varies the release timing, with particles released continuously over a time period. Both methods generate
substantial trajectory spreading that qualitatively resembles the full ensemble variation, though with distinct spatial patterns.

The single-member simulations were performed using velocity fields from individual members of the NATL025-CJMCYC3
ensemble. To ensure robust statistics, we repeated each strategy (spatial and temporal variation) with all $50$ ensemble members
rather than arbitrarily selecting one. For the ensemble simulations, rather than running new simulations where all ensemble
members simultaneously advect particles, we selected and joined trajectories from our existing single-member simulations to
create a 'synthetic' mixture-of-all-member simulation. This mixture simulation contains the full ensemble variability and is
our benchmark for comparing both single-member strategies. The following subsections further detail the two single-member
release strategies and the ensemble simulations, which we refer to as mixture simulations.

## 2.2 Spatially Varying Release

We performed Lagrangian simulations by releasing a cloud of particles around $(35.71°\text{N}, 73.61°\text{W})$, at $1\,\text{meter}$ depth, on
2 January 2010. We evenly spaced the particles in concentric rings around the coordinates, where each ring was placed at a
constant radial separation $(\delta_r)$ from the prior ring, forming a circle of particles. We varied the radius of this cloud of particles;
the larger the radius, the less correlated the velocity vectors of the particles are expected to be, creating more variability in the
trajectories. We created three sets of simulations, with $50$ simulations per set (one per ensemble member). The three sets of
simulations were performed with 7,500 particles, with an initial cloud varying $\delta_r \in \{0.1°, 1.0°, 2.0°\}$.

At the release point, the initial cloud radiuses are approximately $9\,\text{km}$, $90\,\text{km}$, and $180\,\text{km}$. As a reference, we computed the
ensemble average spatial autocorrelation function of the initial particle velocities at the release location on the same release
day (2 January 2010). The spatial autocorrelation function describes the average agreement between the particle velocities of
particles separated by a distance $L$. The larger the separation distance $L$, the more likely their velocities will be decorrelated
(LaCasce, 2008). Assuming that the spatial correlation decays exponentially, we defined the decorrelation length scale $L_L$ as
the e-folding length scale of the exponential that describes the autocorrelations functions (Xia et al., 2013). The analysis is
shown in Appendix A and Figue A1B.

In this region, the average decorrelation length scale for the NATL025-CJMCYC3 ensemble is $L_L = 0.41°$, approximately
$37\,\text{km}$. This $L_L$ is slightly larger than the local Rossby deformation radius, approximately $30\,\text{km}$ in this region (Chelton et al.,
1998). Both spatial scales indicate that the velocities of all the particles released from an initial cloud of $\delta_r = 0.1°$ should be
correlated, while for the larger clouds $\delta_r \in \{1.0°, 2.0°\}$, only a fraction of the particle velocities may be correlated, leading to
more variability in the trajectories.





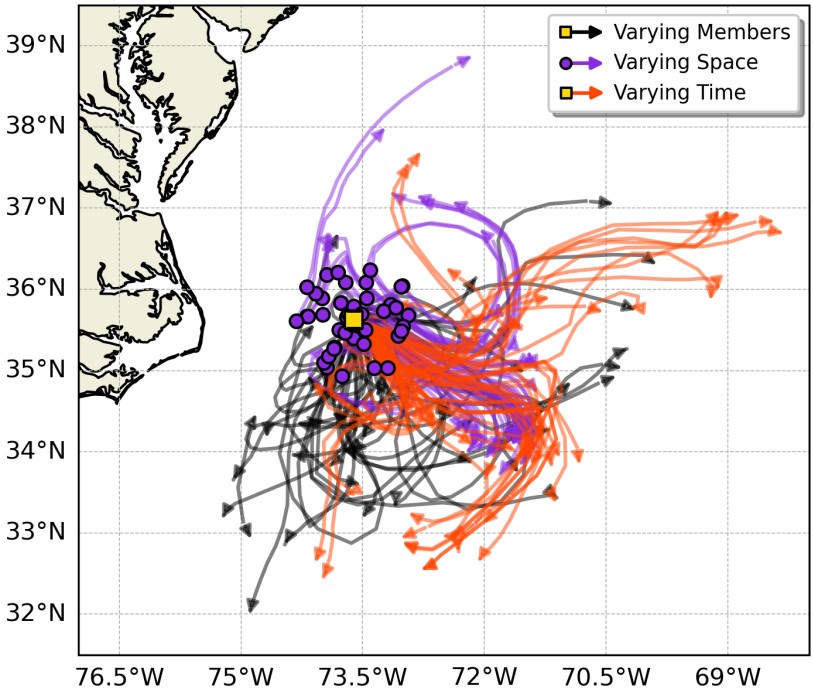

**Figure 1.** Schematic representation of the experiment design, east of Cape Hatteras, showing three approaches to generate variability in the particle trajectories. The black lines show 50 trajectories of particles released from a single point (35.61°N, 73.61°W; yellow square) and advected using velocity fields from all 50 members of the NATL025-CJMCYC3 ensemble. Purple trajectories show 50 randomly selected particles, out of 7,500, released from spatially varying locations (purple circles) within a $1°$ radius of the central point, all advected using ensemble member 3. Orange trajectories represent 50 randomly selected particles, out of 7,500, released uniformly over a 20-week period from the central point (35.61°N, 73.61°W; yellow square), also using ensemble member 3. All trajectories are shown 14 days after their respective release times.

## 2.3 Temporally Varying Release

We also created variability by releasing particles from the same location $(35.71°\text{N}, 73.61°\text{W})$ at different times. We tested three release time windows: 4, 12, and 20 weeks, all starting from 2 January 2010. For each window length, we performed 50 simulations (one per ensemble member), with each simulation releasing 7,500 particles. Within each time window, we distributed the 7,500 particles evenly across the days, resulting in multiple particles being released each day. To ensure particles released on the same day followed different trajectories, we added small random perturbations to their release locations using uniform noise with an amplitude of $0.01°$. We kept this noise amplitude small because larger values would introduce significant spatial variability, making it difficult to isolate the effects of the temporal release strategy alone.

We computed the average decorrelation timescale for all ensemble members to better understand how the particles' initial velocities are correlated for different time lags at the release location. Similar to spatial autocorrelation, the temporal autocorrelation timescale describes the average agreement between the velocities of particles at the same location but with a delay



or lag of $t$ days. The longer $t$, the more likely the velocities will be decorrelated (LaCasce, 2008). Assuming that temporal autocorrelation decays exponentially, we defined the decorrelation timescale $\tau_L$ as the e-folding timescale, after which there is a 69% probability for the velocities to be decorrelated (Xia et al., 2013). The full analysis is shown in Appendix A and

Figure A1C.

We found that the local average decorrelation timescale was $\tau_L = 41$ days, almost 6 weeks. Therefore, it is expected that almost all the particles are correlated for a release period of 4 weeks, and for the larger release periods of 12 and 20 weeks, only a fraction of the particles will be correlated, creating more variability in the trajectories.

## 2.4    Domain Partition and Two-Dimensional Probability Distributions

For the analysis, we created probability distributions from two-dimensional histograms of the positions of particles (Van Sebille et al., 2018). To construct the two-dimensional histograms, we partitioned the domain into hexagonal bins by using the H3 Uber hexagonal hierarchical spatial indexing system (Brodsky, 2018). The H3 grid has the advantage that the area of the hexagons is better preserved across the low and high latitudes compared to a square grid in a Mercator projection (O'Malley et al., 2021; Manral et al., 2023). Additionally, each bin is uniquely indexed, facilitating the reproduction of the analysis. We

used a resolution of $h = 3$ for the hexagons, where the distance between the centroids of two neighboring hexagons measures approximately $100\,\mathrm{km}$. We acknowledge that using a square grid projection for the analysis presented here will not significantly change the results if particles do not drift to high latitudes.

With the hexagonal domain partitioning, we constructed a time series of histograms, where we binned the positions of the particles by counting the number of particles in each bin at timesteps of 1 day. We binned the particle trajectories according to

their particle age, which is defined as the drift time since release. For each day in the time series, we created a two-dimensional probability distribution by normalizing the number of particles in each bin by the total number of particles in the domain on a specific day. The probability distributions, or likelihood distributions, indicate the most likely bins where there are particles at a certain particle age (Pierard et al., 2022). This distribution allows us to define the conditional probability $P_m = P(X|m, t)$ of finding particles in the domain given the ensemble member $m$ used to advect the particles at a particular particle age $t$.

## 2.5    Mixture Probability Distributions

To evaluate how well single-member strategies can reproduce the full ensemble variability, we constructed mixture probability distributions that capture the dispersal patterns across all ensemble members. Using a bootstrapping approach, we randomly selected 150 particles from each of the 50 members and combined their trajectories to create a mixture simulation. We repeated this procedure 50 times to generate a robust set of mixture simulations. From these simulations, we computed mixture prob-

ability distributions $P_{mix}$ by binning particle positions in a hexagonal grid. To assess the effectiveness of our single-member strategies, we computed mixture distributions for each release strategy: three spatial variations ($\delta_r \in \{0.1°, 1.0°, 2.0°\}$) and three temporal variations (4 weeks, 12, and 20 weeks). Computing separate mixture distributions for each strategy was necessary because we could not predict a priori how the spatial or temporal release variations might affect the ensemble variability represented in these distributions.





The optimal number of particles per ensemble member was determined by analyzing the convergence of the distribution's entropy. For our chosen grid cell resolution ($h = 3$), we demonstrate in Appendix B and Figure B1 that the entropy converges with 150 particles per ensemble member, with additional particles providing no significant change in entropy. Therefore, each mixture simulation subset comprises 7,500 trajectories (150 particles $\times$ 50 members). We maintained this total particle count (7,500) in our single-member simulations, both for spatial and temporal release strategies, to ensure direct comparability

between mixture and single-member distributions.

## 2.6    Connectivity Analysis

The connectivity between regions is a useful and powerful analysis performed with Lagrangian simulations (Rypina et al., 2011; Rühs et al., 2013), assessing how many particles originating from one region enter other regions. Within this analysis, we explored if the number of particles reaching each region differs significantly when using mixture simulations instead

of single-member simulations. We also compared how connectivity patterns vary across different mixture strategies (spatial variations with $\delta_r \in \{0.1°, 1.0°, 2.0°\}$ and temporal variations of 4, 12, and 20 weeks). Additionally, we investigated whether single-member simulations with spatially and temporally varying release strategies can reproduce the connectivity statistics of the mixture distributions.

We focused on the connectivity between the surface of the Gulf Stream and the region east of $40°$W. The $40°$W longitude

defines the easternmost boundary where the near-surface waters from the Labrador Current join the Gulf Stream to form the North Atlantic Current (Buckley and Marshall, 2016). This limit also assesses how many particles cross to the easternmost side of the subtropical gyre when released from the surface of the Gulf Stream. In Appendix C, we see this limit in maps showing all places particles drifted to during the six years of simulations. In Figure C1, we present particle dispersion maps for each of the six release strategies (three spatial and three temporal variations) across all 50 ensemble members. Figure C2

shows corresponding dispersion patterns for the 50 subsets of mixture simulations, allowing direct comparison between single-member and mixture approaches. We compared how many particles crossed the $40°$W longitude from the surface of the Gulf Stream in a simulation period of 6 years. We also measured the median time that it took particles to cross $40°$W and the depth at which the particles cross $40°$W.

## 2.7    Marginal Entropy and Relative Entropy Calculation

To compare the dispersion patterns between ensemble members, we took an information theory approach, similar to Cerbus and Goldburg (2013), where we treat each probability distribution as a message. Here, the bins represent the 'alphabet,' and the occurrence of the particles in each bin makes the message, with a probability given by $P$. Each bin $x_i$ contains $\log_2(1/P(x_i))$ information, where $P(x_i)$ is the probability of a character or outcome occurring in a message. The less probable the outcome, the more information it contains; therefore, the less redundant it is. The information can be thought of as the optimal 'length'

that the bin $x_i$ has to be encoded to transmit the message, costing the least amount of bits. Shannon (1948) developed this into a theory of communications in which the fundamental problem is reproducing at one point either exactly or approximately a message selected at another point transmitted over a noisy channel. In this theory, each probability distribution contains



an average amount of information measured by the entropy. The marginal entropy, $H$, measures the intricacy or randomness contained in a distribution and measures the average information content of the distribution (Cover, 1999). The marginal

entropy for the probability distribution is defined as

$$H(X,t) = \sum_{i=1}^{n} P(x_i,t) \log_2 \frac{1}{P(x_i,t)}, \tag{1}$$

where $X$ is the ensemble of bins $x_i$ of the grid, $P$ is the probability distribution associated with the grid, $n$ is the number of bins in $X$, and $t$ is the particle age of the distribution. Marginal entropy measures the minimum number of bits to which the distribution can be compressed or encoded. A distribution with 'more' randomness has less redundancy; therefore, its entropy

is higher. This definition of entropy is equivalent to the definition of entropy in statistical thermodynamics, where entropy is a measure of the number of possible microstates or possible configurations of the system (Shannon, 1948; Cover, 1999). Thus, we define the variability in the dispersal of particles of a simulation as the marginal entropy of its corresponding probability distribution.

The marginal entropy measures the variability of a distribution, but it does not measure how well two distributions match

bin by bin. As illustrated by Olah (2015), consider two probability distributions $P_A(X) = (1/2, 1/4, 1/8, 1/8)$ and $P_B(X) = (1/8, 1/2, 1/4, 1/8)$, both defined over $X = (x_1, x_2, x_3, x_4)$. Both distributions are different when comparing them element by element, that is, $P_A(x_i) \neq P_B(x_i)$. However, if we compute their marginal entropy, we see that they have the same marginal entropy $H_{P_A}(X) = H_{P_B}(X) = 1.75$ bits. Hence, while two distributions may have equivalent marginal entropies, this does not imply that the distributions are equivalent or similar.

Cross-entropy and relative entropy provide better measures for quantifying the difference between two distributions. The cross-entropy measures the average amount of information of a distribution $Q(X,t)$ compared to a reference distribution $P(X,t)$. It is defined as

$$H_P(Q,t) = \sum_{i=1}^{n} Q(x_i,t) \log_2 \frac{1}{P(x_i,t)}, \tag{2}$$

where each bin probability $Q(x_i,t)$ is weighted with the information of the reference distribution $P(x_i,t)$, summed over all

bins $x_i$ at time $t$. The cross-entropy tells us the average information content of $Q$ using the encoding of $P$. From the previous example, the cross-entropy of $P_A$ with respect to $P_B$, or $H_{P_B}(P_A) = 2.25$ bits is larger than its marginal entropy $H(Q)$. Therefore, if we would send messages described by $Q$ with $P$'s encoding, it would be $0.5$ bits more expensive than using its own encoding. The difference between the cross entropy and the marginal entropy is called the relative entropy or Kullback-Leibler Divergence (Kullback and Leibler, 1951) and is defined as

$$D(Q||P,t) = H_P(Q,t) - H(Q,t), \tag{3}$$

where $H_P(Q,t)$ is the cross-entropy of $Q$ with respect to $P$, minus the marginal entropy of $Q$. Eq. (3) is equivalent to the most common definition (Cover, 1999; MacKay, 2003):

$$D(Q||P,t) = \sum_{i=1}^{n} Q(x_i,t) \log_2 \frac{Q(x_i,t)}{P(x_i,t)}. \tag{4}$$



The relative entropy measures the cost of assuming that the distribution is $Q$ when the true distribution is $P$ (Cover, 1999)
and is used to quantify the uncertainty between two distributions.

One of the objectives of this study is to quantify the difference between the mixture distributions $P_{mix}$ and single-member distributions $P_m$, where the variability is created following spatial and temporal release patterns. Given the sparsity of the trajectories sampling the domain, computing the relative entropy between the distributions $P_{mix}$ and $P_m$ implies comparing two-dimensional distributions with zeros in most of the domain. Figure 2A and Figure 2B illustrate this by showing $P_{mix}$
and $P_m$ at a particle age of $t = 15$ days. We see that the probability of finding particles is non-zero in a localized area for both distributions. Therefore, when computing the relative entropy for some bins, it is unavoidable to have terms in which $q \log_2(q/p) \to \infty$ as $p \to 0$. To numerically represent the infinity and compute the relative entropy, we replaced the zeros with a double precision machine epsilon in $P_m$ and $P_{mix}$. The machine epsilon ($\epsilon$) is the smallest number that a computer can represent. For double precision, it is equivalent to $\epsilon = 2^{-52}$, so that the information content of $p = \epsilon$ is equal to $\log_2(1/\epsilon) =$
52 bits.

The relative entropy is non-symmetric, $D(Q||P) \neq D(P||Q)$, and the order in which we compare distributions is crucial. In this study, we calculated the relative entropy as

$$D(P_{mix}||P_m, t) = \sum_{i=1}^{n} P_{mix}(x_i, t) \log_2 \frac{P_{mix}(x_i, t)}{P_m(x_i, t)}, \tag{5}$$

where $P_{mix}$ is the full probabilistic model we aim to reproduce with $P_m$, the reduced-order approximate model computed from
a single member. The relative entropy is computed for the particle age $t$ of the probability distribution. The relative entropy can be interpreted as total information loss (or lack of information) when representing $P_{mix}$ with $P_m$ (Chen et al., 2024; Kleeman, 2002). Figure 2C illustrates computing $D(P_{mix}||P_m, t)$ with the distributions shown in Figures 2A and 2B, where each bin shows the 'information loss', $p_{mix} \log_2(p_{mix}/p_m)$. We note that the bins with information loss coincide with the bins where $P_m$ fails to have particles, but $P_{mix}$ does have particles. Conversely, there is no information loss in bins where there
are no particles for $P_{mix}$, but there are for $P_m$. Therefore, $P_m$ having more bins with particles than $P_{mix}$ is not quantified as information loss. This is more evident when computing $D(P_m||P_{mix}, t)$, in Figure 2D. In contrast, there is information loss in the bins where both distributions have particles but not the same number. There is no information loss if the bins have the same number of particles. By summing over all the bins in $D(P_{mix}||P_m, t)$, we obtain a single value that quantifies the total information loss between the two distributions.

Figure 2D illustrates the opposite case, computing $D(P_m||P_{mix}, t)$ in which the relative entropy measures how well $P_{mix}$ approximates $P_m$. In this case, there is only information loss in the bins where $P_m$ and $P_{mix}$ have particles, although $P_{mix}$ covers more bins. This again shows that there is no information loss for having a wider probability that covers a larger area, containing the bins of the distribution to represent. By summing over all bins in $D(P_m||P_{mix})$, we get a relative entropy of 2.8 bits, which is far less than $D(P_{mix}||P_m, t)$ described previously.

To summarize, because of the asymmetry in the relative entropy, it is important to evaluate the full probabilistic model with the encoding of the reduced-order model, $D(P_{mix}||P_m, t)$, in Eq. (5). In that case, the relative entropy quantifies the uncertainty when using the simplified probabilistic model ($P_m$) to approximate the full model ($P_{mix}$) (Chen et al., 2024).



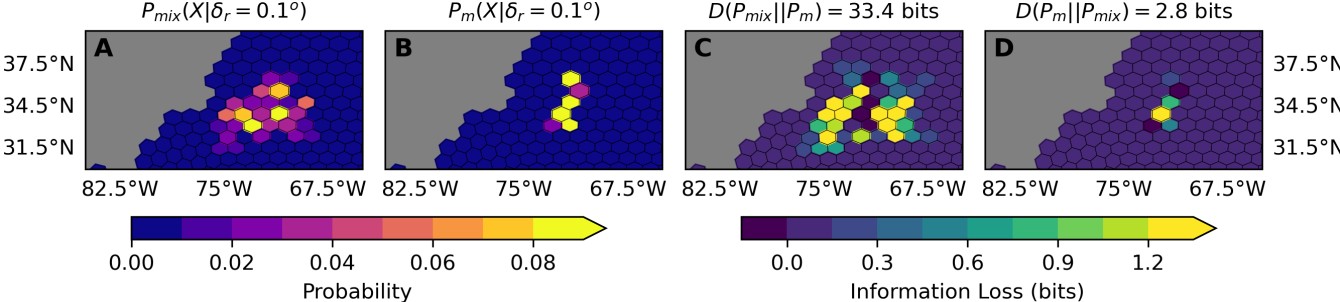

**Figure 2.** Comparison of probability distributions and their relative entropy. (A) Mixture distribution $P_{mix}(X|\delta_r = 0.1°)$ at 15 days after release, representing the full probabilistic model. (B) Single-member distribution $P_m(X|\delta_r = 0.1°)$ at 15 days after release, representing the reduced-order approximate model. (C) Information loss map showing the contribution of each grid cell to the total relative entropy $D(P_{mix}||P_m)$ when approximating the mixture distribution with the single-member distribution. (D) Information loss map showing the contribution of each grid cell to the total relative entropy $D(P_m||P_{mix})$ when approximating the single-member distribution with the mixture distribution. Gray hexagons represent land. Color scales show probability values (A and B) and information loss in bits (C and D). The zero-bit value falls within the second color bin from the left in the information loss color scale.

## 3 Results

### 3.1 Connectivity

This section compares mixture simulations (containing the full ensemble variability) and single-member distributions for particles crossing the $40°$W line. Throughout this analysis, we use the mixture distribution with $\delta_r = 0.1°$ as our reference, as it represents the closest approximation to a point release while still containing the full ensemble variability. This allows us to consistently evaluate how increasing spatial or temporal variability in single-member simulations compares to this baseline case. We employed Empirical Cumulative Distribution Functions (ECDFs) to assess the likelihood of single-member distributions matching the average particle counts in mixture distributions. Figure 3 shows the ECDFs for the number of particles crossing

$40°$W and the median particle age at which they cross that longitude. Figures 3A and 3B compare spatially varying releases, whereas Figures 3C and 3D compare temporally varying release simulations. In all panels, the ECDF curves represent the single-member distributions, and the vertically shaded lines show the $99\,\%$ confidence interval of their corresponding mixture distributions. The mixture distributions are depicted as vertically shaded lines to enhance the readability of the plots since they

are well-defined Gaussian distributions. The plots showing Kernel Density Estimate (KDE) distributions of the single-member and of the mixture distributions can be found in Figures C3 and C4, in the Appendix C.

     Figure 3A shows greater variance in single-member distributions, with values ranging from $1,000$ to $5,100$ particles, compared to the mixture distributions. This increased variability occurs because single-member distributions reflect the specific ocean conditions of individual ensemble members, while mixture distributions average out these individual variations across

multiple members, resulting in more stable statistics. On average, more particles cross the $40°$W line for simulations with larger release clouds $\delta_r$ in the single-member distributions. The same relation between $\delta_r$ and the number of particles crossing





is observed in the mixture distributions. The ECDF provides insights into the probability of single-member simulations not capturing the mixture distribution averages. For instance, in single-member simulations with a release radius of $\delta_r = 0.1°$, there is a 0.64 probability of having fewer particles crossing the 40°W line than the average of the mixture distribution with

$\delta_r = 0.1°$, and consequently, a 0.36 probability of overestimation. This probability of underestimation decreases to 0.34 (with 0.66 probability of overestimation) for $\delta_r = 1.0°$ and to 0.10 (with 0.90 probability of overestimation) for $\delta_r = 2.0°$, taking the same mixture distribution ($\delta_r = 0.1°$) as reference.

Figure 3C shows the ECDFs for temporally varying releases. The distributions for the single-member simulations with 4, 12, and 20-week releases are similar but show more variance than the mixture distributions represented by the shaded lines.

Mixture distributions for 4 and 12-week releases have comparable average particle counts, while 20-week releases show slightly lower averages. For single-member simulations, the probability of having fewer particles than the mixture distribution average ($\delta_r = 0.1°$) is 0.56 (with 0.44 probability of overestimation) for 4-week releases, 0.50 (with 0.50 probability of overestimation) for 12-week releases, and 0.66 (with 0.34 probability of overestimation) for 20-week release periods.

Figure 3B shows the ECDFs for the median particle age of particles crossing 40°W in spatially varying release simulations.

The single-member distributions (ECDF curves) show a clear separation based on the release cloud size ($\delta_r$). Particles from smaller release clouds ($\delta_r = 0.1°$) tend to have longer median drift times, while those from larger release clouds ($\delta_r = 2.0°$) have shorter median drift times. This trend is also reflected in the mixture distributions' 99 % confidence interval (shaded lines). While the single-member simulations show a greater spread in median drift times compared to the mixture distributions, they maintain the same general pattern of decreasing drift times with increasing release cloud size. However, the wider spread

in single-member distributions indicates that individual simulations may not consistently reproduce the more stable statistics captured by the mixture distributions.

Figure 3D shows the ECDFs for particle age in temporally varying release simulations. The distributions for different release durations (4, 12, and 20 weeks) are more closely aligned than the spatial variations in panel B. However, longer release periods (20 weeks) tend to show slightly shorter median drift times. While single-member distributions still exhibit greater variability

than the mixture distributions, this variability is less pronounced than in the spatially varying simulations. This suggests that temporal release variations may provide more consistent reproducibility of mixture statistics compared to spatial variations, although this varies in individual simulation results.

In summary, our connectivity analysis reveals that single-member simulations tend to either significantly under- or overestimate particle transport across 40°W, with the bias depending on the release strategy. For spatial variations, larger release clouds

($\delta_r = 2.0°$) show a strong tendency to overestimate connectivity (90 % probability), while smaller release clouds ($\delta_r = 0.1°$) are more likely to underestimate it (64 % probability). Temporal variations show more balanced probabilities of under- and overestimation, particularly for 12-week releases (50-50% probability), and generally exhibit less pronounced variability in particle ages compared to spatial variations.




**Figure 3.** Connectivity analysis between the Gulf Stream at Cape Hatteras and the line at $40°$W in the North Atlantic. The plots compare single-member ECDFs (lines) with mixture distribution average plus/minus $99\,\%$ confidence values (shaded vertical lines). A) ECDFs of the number of particles crossing the line for spatially varying simulations. B) ECDFs of the median particle age distributions for spatially varying releases. C) ECDFs of the number of particles from temporally varying simulations. D) ECDFs of the median particle age distributions for temporally varying simulations.





## 3.2    Marginal and Relative Entropy

We calculated the marginal entropy, Eq. (1), for every single-member and corresponding mixture distribution to assess the variability and determine which release strategies can represent the variability of the full ensemble. In total, we computed the marginal entropy functions for all six sets of single-member distributions and the six sets of mixture distributions. Each set had 50 distributions. For each set, we calculated the average and the standard deviation of the marginal entropy functions, resulting in one entropy curve as a function of particle age per set. Figure 4A illustrates the average entropy curves for spatially varying

release distributions, while Figure 4B shows those for temporally varying release distributions. Detailed entropy curves for each single member and mixture simulation are provided in Figures C5 and C6 in the Appendix C.

Figure 4A shows the marginal entropy as a function of particle age for various spatial release strategies, comparing single-member probability distributions ($P_m$) with mixture distributions ($P_{mix}$) using different spatial release intervals ($\delta_r$). Three single-member curves are shown: $\delta_r = 0.1°$ (blue dotted line), $\delta_r = 1.0°$ (purple dashed line), and $\delta_r = 2.0°$ (green dash-dot

line). Two mixture distribution curves are presented: $\delta_r = 0.1°$ (solid black line) and $\delta_r = 2.0°$ (black dash-dot line). All curves show a logarithmic increasing trend in entropy with particle age, indicating growing dispersion over time. The single-member distributions with larger $\delta_r$ values ($1.0°$ and $2.0°$) initially overestimate the entropy compared to the mixture distribution with $\delta_r = 0.1°$, particularly in the first 10 days. After this period, only the single-member distribution with $\delta_r = 2.0°$ adequately represents the variability of the mixture with $\delta_r = 0.1°$. Shaded areas around the single-member curves represent the standard

deviation, illustrating the spread of entropy values across the ensemble. There is no shaded area around the mixture entropy curves because their standard deviation was of the order of magnitude $10^{-2}$ bits. The logarithmic scale on the $x$-axis emphasizes the rapid changes in entropy during the early stages of particle dispersion.

Figure 4B shows the entropy as a function of time for the temporal varying release strategies and their corresponding mixture distributions, comparing single-member probability distributions $P_m$ for different release periods against mixture

distributions ($P_{mix}$). The single member distributions are shown for release periods of 4, 12, and 20 weeks. These curves show a general trend of entropy increasing logarithmically over time, with longer release periods resulting in higher entropy values. Two mixture distributions are plotted: one subsampled from a 4 week release and another subsampled for $\delta_r = 0.1°$. We compared temporal and spatial mixture distributions to understand how different release strategies contribute to the total ensemble variability. These mixture distributions consistently show higher entropy values than single-member distributions,

indicating that $P_m$ captures less variability than the mixture distributions. The 20 week single member distributions closely follow the mixture distribution with $\delta_r = 0.1°$, often overlapping or slightly exceeding it. Among the single member curves, the 20 week release generally shows the highest entropy, followed by 12 and 4 weeks in descending order. However, these differences become less pronounced as time increases.

Comparing spatial and temporal strategies in Figures 4A and 4B, we establish the mixture distribution with $\delta_r = 0.1°$ as our

reference standard, as it shows the minimum variability among all mixture strategies. The 20-week single-member distributions most closely approximate this reference, while the single-member spatial releases show more variable performance. Both the $\delta_r = 2.0°$ and the 20-week mixture distributions exhibit the highest entropy values, demonstrating how combining either





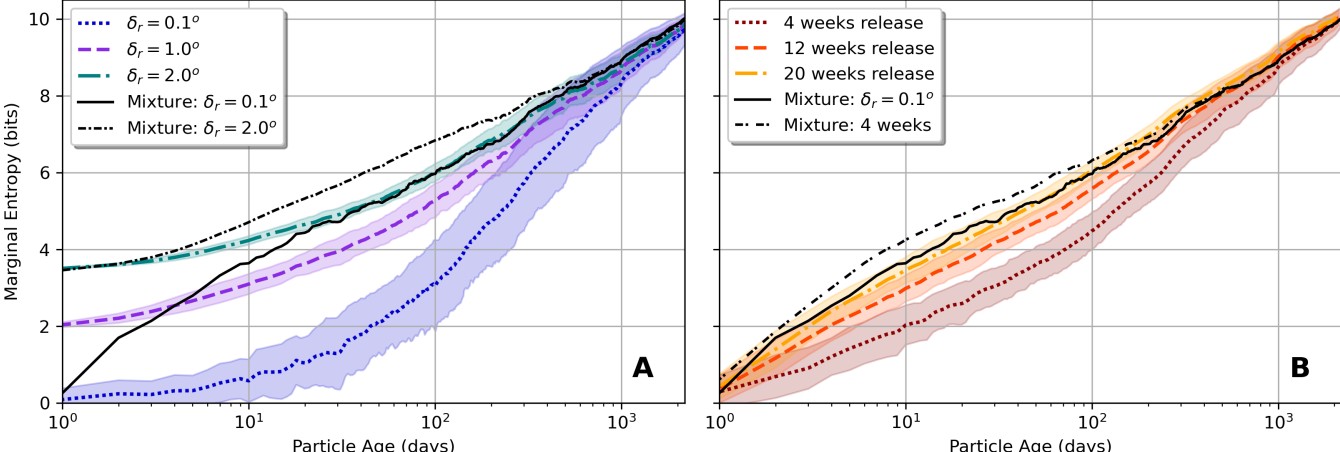

**Figure 4.** Average marginal entropy as a function of particle age of single-member distributions (colored lines) and mixture distributions (black lines). The shaded areas represent the standard deviation. The particle age is on a logarithmic scale. A) Comparison of the single-member and mixture distributions with spatially varying release. B) Comparison of the single-member and mixture distributions with temporally varying release. In both panels, we added the entropy curve of the mixture $\delta_r = 0.1°$ as a reference.

spatial or temporal release variability with ensemble variability increases the total dispersion. This reinforces our choice of the more point-like $\delta_r = 0.1°$ mixture as our reference for evaluating single-member approximations. For clarity, we omitted the intermediate mixture curves ($\delta_r = 1.0°$, 12-week, and 20-week), as their entropy values consistently fall between those of the $\delta_r = 0.1°$ and $\delta_r = 2.0°$ distributions.

We computed the relative entropy as a function of particle age, Eq. (5), by comparing single-member distributions with mixture distributions. Also, we computed the relative entropy of every ensemble member with every mixture distribution individually. Therefore, we calculated $50^2 = 2,500$ relative entropies when comparing two sets of distributions. We computed the average and the standard deviation of the 2,500 relative entropy functions for each of the six sets of single member distributions ($\delta_r \in \{0.1°, 1.0°, 2.0°\}$ and 4, 12 and 20 weeks), relative to the six sets of mixture distributions, ending up with 36 average relative entropy functions.

Figure 5 shows the average relative entropy as a function of particle age, divided into four panels (A-D), each using a different mixture distribution as a reference. We omitted the plots where the mixture for the $\delta_r = 1.0°$ and 12-weeks strategy as a reference since their entropy lies between their corresponding extreme strategies, that is $\delta_r \in \{0.1°, 2.0°\}$, and 4 and 20-weeks releases. In all panels, the dotted and dashed lines represent the average relative entropy for each strategy, while the shaded areas around these lines indicate the standard deviation. The standard deviation measures the variability in relative entropy, revealing that extreme cases exist where single-member distributions represent the reference mixture distributions poorly.

Figure 5A uses the mixture distribution with $\delta_r = 0.1°$ as the reference. On average, the single-member distribution with $\delta_r = 2.0°$ (green dotted line) most closely approximates the reference mixture, having the lowest mean relative entropy across



most of the time range. The 20-week release strategy (red dashed line) performs similarly well. However, the large standard deviations, particularly for $\delta_r = 0.1°$ (blue dash-dot line) and the 4-week strategy (orange dotted line), indicate significant variability in how well these strategies represent the reference mixture, indicating a greater lack of information.

Figure 5B, referencing the mixture distribution with $\delta_r = 2.0°$, shows that the single-member $\delta_r = 2.0°$ strategy most closely matches this reference mixture on average. The 20-week release strategy also performs well, especially for higher particle ages. However, this 20-week release strategy shows a distinctly different evolution pattern with a constant decrease in relative entropy compared to other strategies. The substantial standard deviations for all strategies, particularly pronounced for $\delta_r = 0.1°$ and the 4-week strategy, highlight the potential for large discrepancies between individual simulations and the reference mixture.

Figures 5C and 5D use the 4-week and 20-week mixture distributions as references, respectively. The corresponding single-member temporal release strategies show the lowest mean relative entropy in both cases. However, the wide standard deviation bands, particularly noticeable for the spatial release strategies ($\delta_r = 0.1°$ and $\delta_r = 2.0°$), underscore the high variability in how well these strategies capture the reference mixture's characteristics.

Across all panels of Figure 5, relative entropy peaks between 10 and 100 days of particle age, with the largest standard
deviations also occurring in this range. Notably, standard deviations for temporal (20-week) and spatial ($\delta_r = 2.0°$) strategies peak at different times: the 20-week release shows maximum variability at earlier particle ages, while the $\delta_r = 2.0°$ release peaks later. This suggests that single-member distributions are most likely to significantly diverge from the mixture distributions during this time period. Figures C7 and C8, in the Appendix C, illustrate this variability of two randomly selected probability distributions of different release strategies at particle ages of 10, 100 and 1,000 days. These figures also show randomly selected
subsets of the mixture distributions at the same particle age.

From the average entropy curves shown in Figure 5, we took the average over the 6 years the particles were drifting after release. We compiled these values for the 36 comparisons between mixture and single-member sets, with different release strategies in Figure 6. This figure presents a heatmap of the time-averaged relative entropy values for various combinations of single-member and mixture distributions. The rows represent single-member distributions, while the columns represent mixture
distributions. The color scale ranges from dark green (lowest relative entropy) to light green (highest relative entropy), with numerical values provided in each cell. Notably, the 20-week single-member distribution (bottom row) consistently shows the lowest relative entropy across all mixture distributions, indicating it best represents the ensemble variability. Conversely, the $\delta_r = 0.1°$ single-member distribution exhibits the highest relative entropy values, suggesting it is the least effective at capturing the characteristics of the mixture distributions.

## 4    Discussion and Conclusions

In this study, we investigated how to generate ensemble-like variability within single-member Lagrangian simulations by implementing varying spatial and temporal release strategies in the Gulf Stream region near Cape Hatteras. The surface connectivity between the Gulf Stream and the region past $40°W$ revealed significant differences in the number of particles crossing between different release strategies in the single-member distributions. The ECDFs in Figure 3 showed that, for spatially varying





**Figure 5.** Relative entropy as a function of particle age for different single-member distributions compared to mixture distributions. The subplots A-D show comparisons using different reference mixture distributions: (A) $\delta_r = 0.1°$, (B) $\delta_r = 2.0°$, (C) 4-week release, and (D) 20-week release. Dotted and dashed lines represent the average relative entropy for each strategy, while shaded areas indicate standard deviation. The x-axis shows particle age in days (log scale), and the y-axis shows relative entropy in bits. Different colors represent various single-member strategies: $\delta_r = 0.1°$ (blue), $\delta_r = 2.0°$ (green), 4-week release (red), and 20-week release (orange).



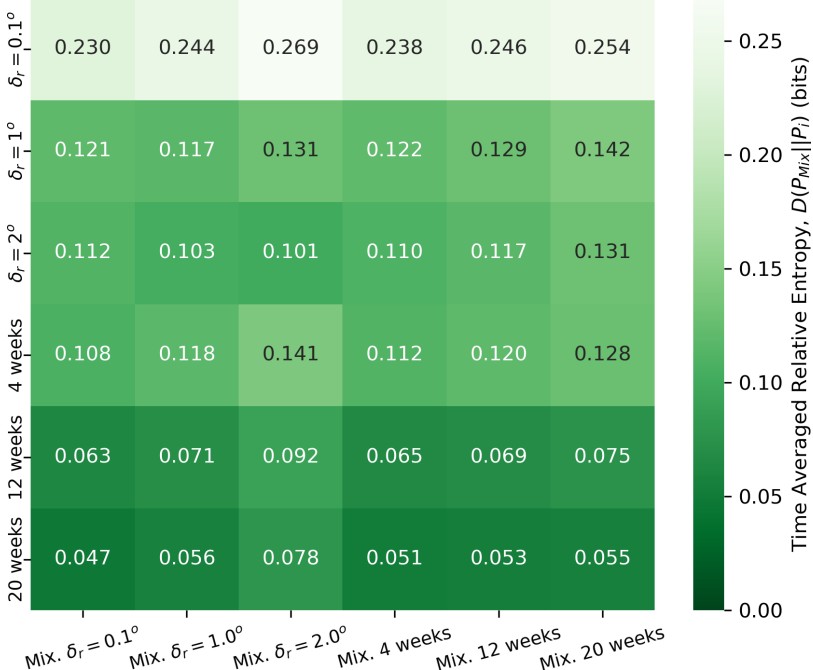

**Figure 6.** Time-averaged relative entropy (in bits) between single-member and mixture distributions for different release strategies. Rows represent single-member distributions, and the columns represent mixture distributions. Color intensity indicates the magnitude of relative entropy, with dark green representing lower values (better agreement) and light green representing higher values (poorer agreement). Numerical values in each cell show the precise time-averaged relative entropy.

releases, the larger the initial particle cloud, the more particles cross the $40°$W. Regarding the temporal distributions, we did not see significant variations in the number of particles crossing $40°$W; the distributions for the number of particles and the median times were similar between the three temporal release strategies. Moreover, the normal distribution observed in the mixture distributions can be attributed to the central limit theorem. This fundamental principle in probability theory states that when independent random samples are drawn from a population with a finite variance, the distribution of their means will approximate a normal distribution as the sample size increases. In our case, the bootstrapping method used to construct the mixture distributions effectively simulates this sampling process, resulting in the observed normal distributions.

Regarding representing the full ensemble variability with single-member simulations in the connectivity analysis, we see that particles are more consistent in crossing the $40°$W meridian in the mixture distributions. Therefore, when comparing mixture distributions with single-member distributions, we counted the percentage of single-member simulations with fewer particle crossings than the mixture distribution with $\delta_r = 0.1°$. In this analysis, we saw that performing a one-time spatial release with a radius of $\delta_r = 2.0°$ better represents the particle crossings in the mixture distributions. From all the release strategies, single-member simulations with $\delta_r = 2.0°$ release cloud had the lowest likelihood of having fewer particles crossing than the mixture simulation with $\delta_r = 0.1°$. This might be because a large initial cloud of particles releases more particles outside the Gulf



Stream, creating a wider variety of trajectories that cross $40°$W. In the case of the temporal releases, the single member with
20-week releases had fewer particles crossing $40°$W, with a 66% likelihood of having fewer particles than the mixture with
$\delta_r = 0.1°$. This likelihood is 10% higher than 4 and 12-week single-member distributions, suggesting that the seasonability
may be playing a role in the transport of particles to the eastern side of the domain. The connectivity with the eastern region
of the domain might be stronger during winter, corresponding to the release of particles in the 4-week period (from 2nd to 30
of January) and 12-week period (from 2 January to 27 March). Meanwhile, the 20-week release, from 2nd January to 22 May,
had a portion of its particles released during spring.

The marginal entropy analysis, shown in Figure 4, provided insights into how well different release strategies represent the
full ensemble variability. In general, we saw how the marginal entropy increased with time for all strategies considered, some
at slower rates than others. We attributed this to the percentage of particles released under the local decorrelation length and
times for the different strategies. For instance, spatial releases with radius $\delta_r = 0.1°$ and temporal releases of 4-weeks, which
exhibited the lowest marginal entropy, had all their particles released within their respective decorrelation scales. As the radius
or release period was increased, there were more particles with decorrelated initial velocities, resulting in higher entropy.

It is important to highlight that the marginal entropy of the mixture distributions consistently exceeds that of corresponding
single-member distributions, demonstrating that ensemble simulations under identical release conditions inherently generate
greater trajectory variability than single-member simulations. By maintaining equal particle counts between mixture and single-
member simulations, we ensured that the higher entropy in mixture distributions reflects genuine ensemble dynamics rather
than statistical artifacts. The higher marginal entropy in mixture distributions may also be attributed to the temporal context of
our study: we advected particles $\sim 18$ years after the initialization of the NATL025-CJMCYC3 ensemble (which was perturbed
during 1993). At the release date of the particles (2010), the perturbations had sufficient time to adapt and decorrelate the
velocity fields of the members, which suggests that ensemble Lagrangian dispersion arises not only from mesoscale chaos but
also from low-frequency, large-scale intrinsic fluctuations.

Our analysis compares both spatial and temporal release strategies against the reference mixture simulation with $\delta_r = 0.1°$.
For spatial releases (Figure 4A), we found significant limitations. The larger release areas ($\delta_r = 1.0°$ and $\delta_r = 2.0°$) initially
overestimate variability during the first 10 days, as particles start from a wider area than the reference's mixture with $\delta_r = 0.1°$
radius. While $\delta_r = 2.0°$ simulations eventually match the reference entropy after 30-40 days, $\delta_r = 1.0°$ simulations underes-
timate it until about $1,000$ days after release. In contrast, temporal release strategies (Figure 4B) show better performance,
particularly the 12 and 20-week releases. The 20-week release strategy consistently matches the reference mixture's entropy
across all temporal scales, demonstrating that continuous particle releases over time can effectively reproduce the variabil-
ity captured by ensemble simulations. This suggests that temporal variation in release times is more effective at representing
ensemble variability than increasing the spatial extent of the release area.

As we explained in Section 2.7, two distributions that have the same entropy do not necessarily exhibit the same distributions
since two different probability distributions can have equivalent entropies. We compared the relative entropy to measure the
agreement between two distributions, which measures the lack of information when representing the full ensemble with a
single-member simulation. In this framework, we found that performing an ensemble simulation is more informative than a



single-member simulation. The relative entropy quantifies the lack of information, or in other words, quantifies the uncertainty,
by measuring the agreement between the distributions. The relative entropies, shown in Figure 5, further support the findings
from the marginal entropy assessment. The 20-week release generally showed the lowest relative entropy with respect to
the mixture using $\delta_r = 0.1°$, indicating this release strategy most effectively captured the variability in the trajectories of the
full ensemble. Despite this, the standard deviation of the relative entropy of the 20-week simulations indicated that individual
simulations could deviate substantially for timescales less than 100 days after release. In addition, comparing the time-averaged
relative entropy, shown in Figure 6, showed how 20-week releases have less uncertainty across different reference mixtures,
followed by 12-week releases. On average, a $\delta_r = 2.0°$ and a 4-week release had similar uncertainties compared to all mixture
distributions. This further supports the idea that performing long continuous releases is the best release strategy to represent
the ensemble variability.

In single-member simulations, we demonstrated that releasing particles at slightly different locations or times can match the
variability in the behavior of particles released at a specific time and location from an ensemble of simulations. An interpretation
of this may be that an ensemble of Lagrangian simulations has an ergodic flavor in which statistical homogeneity exists between
an ensemble of simulations and single-member simulations (Shannon, 1948). However, this does not constitute proof of the
system's ergodicity.

While our study provides valuable insights into generating ensemble-like variability in single-member simulations, several
limitations should be acknowledged. Our analysis focused solely on the Gulf Stream region near Cape Hatteras, and the effec-
tiveness of these release strategies may vary in other oceanic regions with different dynamics. Additionally, while our particles
were advected in three-dimensional flows, we only considered surface particle releases, which may not fully represent the three-
dimensional transport processes occurring throughout the water column. Our results are based on the NATL025-CJMCYC3
model configuration, and the effectiveness of these strategies may be resolution-dependent, as higher-resolution models resolve
smaller-scale processes that could introduce additional variability in transport pathways. Furthermore, our study was limited to
forward-in-time simulations, whereas backward-in-time tracking could provide complementary information about generating
ensemble variability in single-member simulations in studies concerning source regions and transport pathways. Future work
should explore the applicability of these methods across different oceanic regions, depths, and temporal directions to establish
more comprehensive guidelines for single-member Lagrangian simulations.

Ensemble simulations remain the standard for capturing the full range of variability in ocean simulations; our study provides
guidance on releasing particles in single-member simulations to increase the variability of the trajectories and, in this case, bet-
ter represent ensemble statistics. While data assimilative models excel at improving mean state predictions through observation
integration, ensemble approaches are better suited for exploring the full range of possible outcomes and quantifying uncertainty
in trajectory predictions. Generating ensemble-like variability for Lagrangian simulations advected using assimilative models
could be particularly powerful: applying spatial or temporal release strategies could help capture both the improved mean state
from data assimilation and the trajectory variability typical of ensemble simulations. These findings have important implica-
tions for ocean modeling and particle tracking studies, especially when computational resources limit the use of full ensemble





simulations. By carefully selecting release strategies, researchers can maximize the variability of single-member simulations,
potentially improving predictions of particle transport by capturing extreme events.

**Appendix A: Spatial and Temporal Autocorrelations at the Release Location**

We computed the spatial and temporal autocorrelation functions of the horizontal velocity vectors at the time and location of
release of the particles. The spatial autocorrelation functions were calculated over a set of points placed over a west-to-east
line, shown by the blue dots in Figure A1A, with a horizontal spacing of $0.01°$. We calculated the autocorrelation function
from these points as a function of the distance $L$. The spatial autocorrelation function is defined as

$$\rho(L) = \left\langle \frac{\mathbf{u}(r_0 + L) \cdot \mathbf{u}(r_0)}{\|\mathbf{u}(r_0 + L)\|\|\mathbf{u}(r_0)\|} \right\rangle, \tag{A1}$$

in which we compute the dot product of a pair of vectors $\mathbf{u}(r_0)$ and $\mathbf{u}(r_0 + L)$, divided by the multiplication of their norms,
averaged over all the pairs of particles (Xia et al., 2013). In Eq. (A1), $\|\cdot\|$ is the usual $L_2$ norm, and $\langle\cdot\rangle$ indicates an average
over particle pairs. We computed $\rho(L)$ for the range $L \in [0.01°, 2.00°]$, with a $0.01°$ spacing. The autocorrelation function
is defined between $[-1, 1]$, in which $\rho(L) = 1$ indicates a full positive correlation, $\rho(L) = -1$ a full negative correlation, and
495 $\rho(L) = 0$ no correlation.

Following Eq. (A1), we computed $\rho(L)$ for each of the 50 ensemble members of the NATL025-CJMCYC3. In Figure A1B,
we show the $\rho(L)$ for each ensemble member as black lines. We see great variability in the curves but an exponentially decaying
trend in which, as L increases, the particle velocities are less correlated. We performed an exponential fit, $e^{x/L_L}$, of the 50
correlation curves, shown in blue in Figure A1B. From the exponential fit, we obtained a decorrelation length $L_L = 0.41°$,
which corresponds to approximately $37\,\mathrm{km}$ at a latitude of $35.5°\mathrm{N}$. As a reference, the Rossby deformation radius in this
region is $L_R \approx 30\,\mathrm{km}$ (Chelton et al., 1998).

Similarly, we computed the temporal autocorrelation functions by sampling the velocity at the same location but on different
days, shown as a red point in Figure A1A. We sampled the velocity daily for a duration of 60 days, starting on the 2nd of
January, 2010. From the sampled velocities, we computed the temporal autocorrelation function given by

$$\rho(t) = \left\langle \frac{\mathbf{u}(t_0 + t) \cdot \mathbf{u}(t_0)}{\|\mathbf{u}(t_0 + t)\|\|\mathbf{u}(t_0)\|} \right\rangle, \tag{A2}$$

where $t$ represents the time lag between pairs of velocities $\mathbf{u}(t_0)$ and $\mathbf{u}(t_0 + t)$ averaged over all pairs with a lag $t$, similar to
Eq. (A1).

Similarly to $\rho(L)$, we computed the temporal autocorrelation function $\rho(t)$ for the 50 members of NATL025-CJMCYC3,
for the range $t \in [1, 60]$ days with a spacing of 1 day. In Figure A1C, we show each member's $\rho(t)$ as black curves. We
performed an exponential fit $e^{x/\tau_L}$ over the 50 correlation curves. In Figure A1C, we show in red the exponential fit. We found
a decorrelation timescale of $\tau_L = 41$ days for the velocities of the particles released on different days.





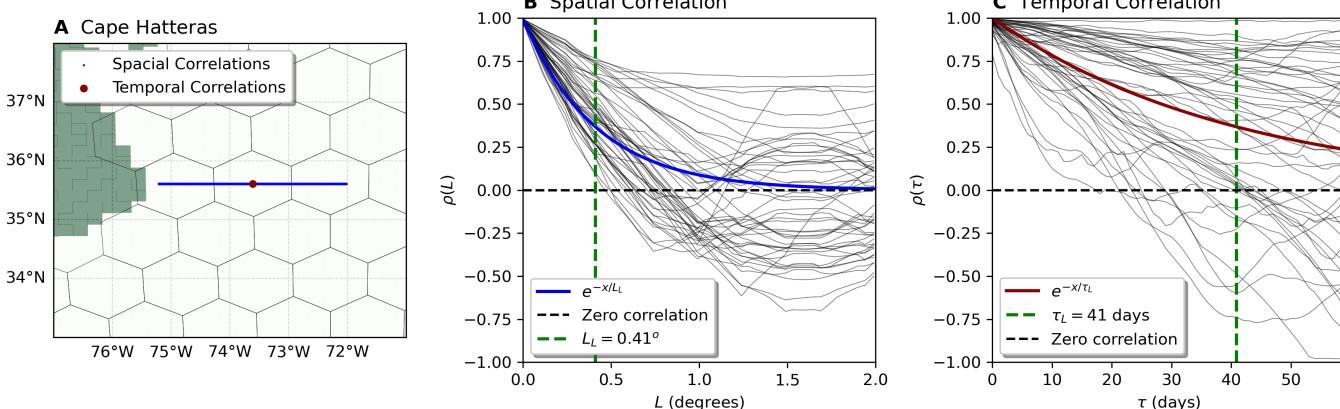

**Figure A1.** A) Map of Cape Hatteras showing the points used to compute the spatial correlations (blue) and the location used to compute the temporal correlations (red). The hexagons mark the limits of the hexagonal grid, and the green area represents the North American coast. B) Spatial correlations function around the release location, and each black line shows the correlation function for an ensemble member. The blue line shows the exponential fit computed over the 50 correlation functions. The green line shows the decorrelation length scale $L_L = 0.41° \approx 37\,\text{km}$. C) Temporal Correlations with velocities sampled daily for 60 days from the 2nd of January 2010. The black lines show the correlation functions of single ensemble members, and the red line shows the exponential fit with a decorrelation timescale of 41 days.





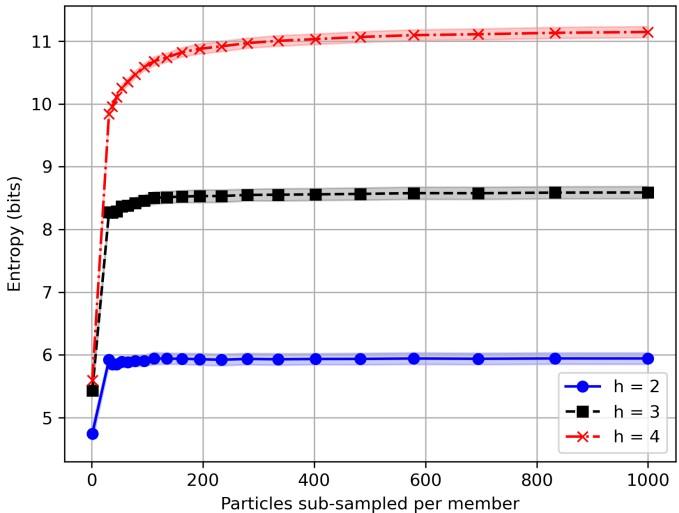

**Figure B1.** Marginal entropy of mixture distributions as a function of particles sampled per ensemble member, shown for three different hexagonal grid resolutions ($h$). Higher $h$ values indicate finer spatial resolution, resulting in higher entropy values.

### Appendix B: Marginal Entropy as a Function of Number of Particles and Grid Resolution

The calculation of the mixture probability distributions ($P_{mix}$) requires determining both the optimal number of particles to sample and the appropriate spatial resolution for binning these particles. These parameters directly affect the entropy of the resulting distributions. We investigated this relationship by varying two key parameters: the number of particles sampled per ensemble member and the hexagonal grid resolution ($h$).

Figure B1 shows how the entropy converges as we increase the number of particles sampled per ensemble member, plotted for three different grid resolutions ($h \in \{2, 3, 4\}$). As expected, finer grid resolutions (larger $h$ values) yield higher entropy values as they capture more detailed spatial information. For our chosen grid resolution of $h = 3$, the entropy converges to approximately $8.5$ bits when sampling 150 or more particles per ensemble member. Coarser resolutions ($h = 2$) require fewer particles to converge, while finer resolutions ($h = 4$) need more particles but capture more spatial detail. Based on this analysis, we selected $h = 3$ and 150 particles per member as sufficient parameters for our study, balancing computational efficiency with spatial resolution.



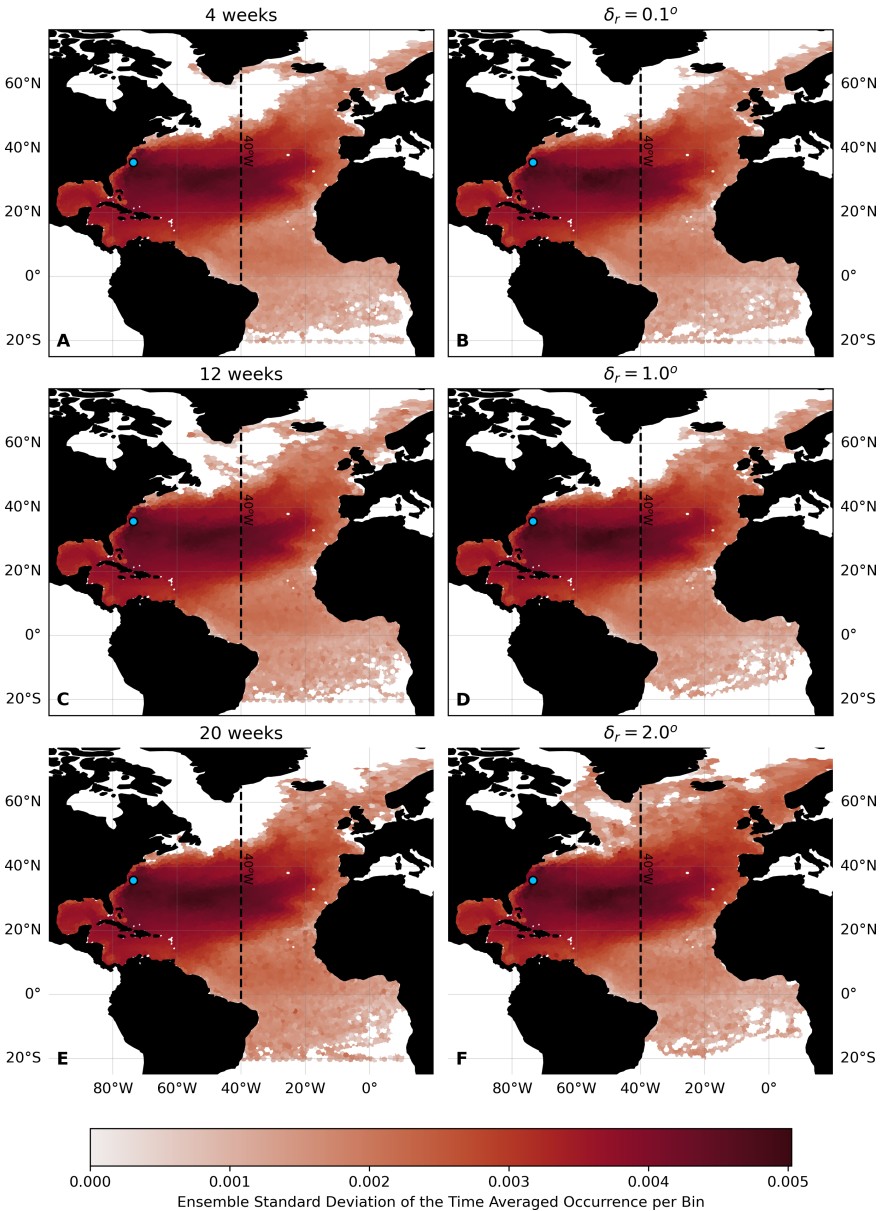

**Figure C1.** Ensemble standard deviation of time-averaged particle occurrence per bin in the North Atlantic Ocean for single-member simulations. Left column (A, C, E): Temporal release strategies at 4 weeks, 12 weeks, and 20 weeks. Right column (B, D, F): Spatial release strategies with $\delta_r \in \{0.1°, 1.0°, 2.0°\}$. The color scale represents the ensemble standard deviation of a 6-year time-averaged occurrence per bin. The maps illustrate the variability in particle dispersal for single-member simulations. The dashed line at $40°$W indicates the eastern boundary of the study area. The blue dot marks the approximate release location.

## Appendix C: Additional Supplementary Figures





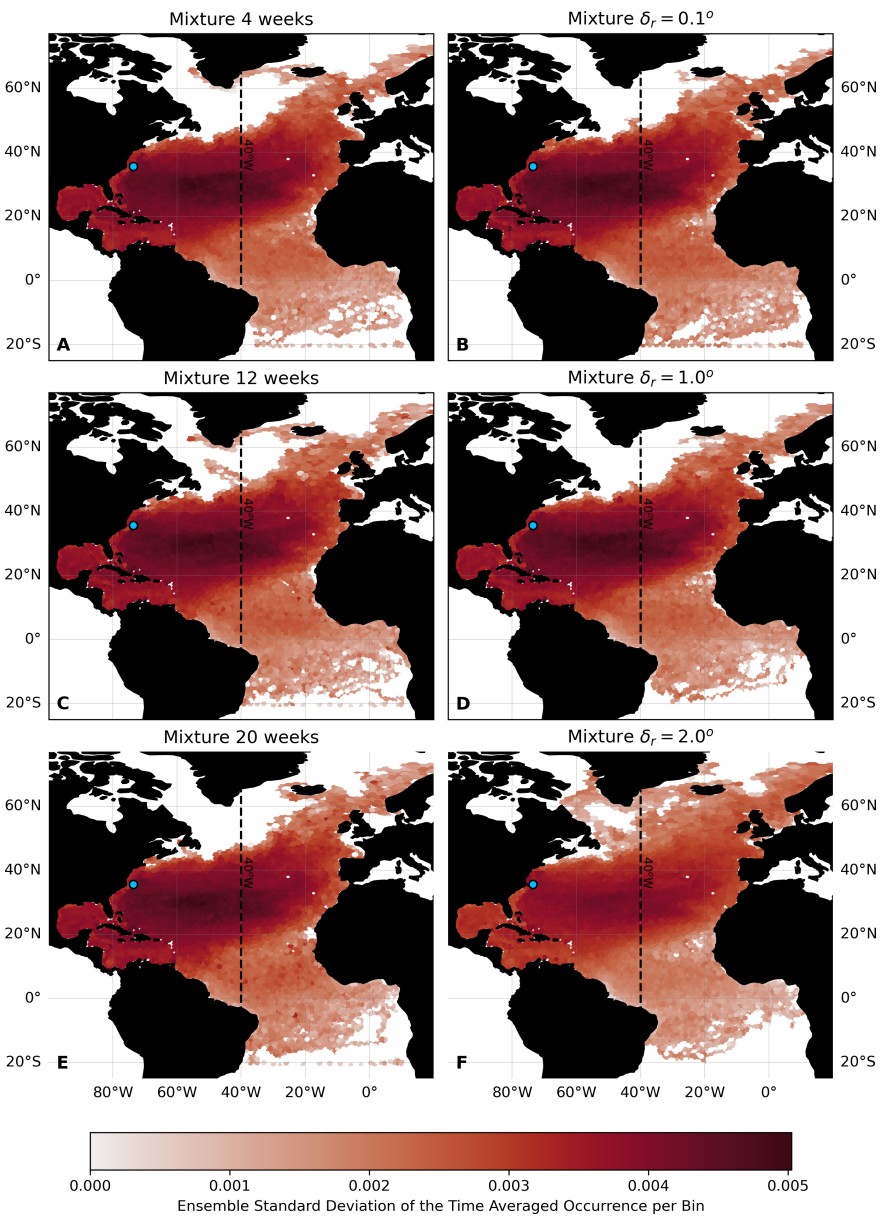

**Figure C2.** Ensemble standard deviation of time-averaged particle occurrence per bin in the North Atlantic Ocean for mixture simulation subsets. Left column (A, C, E): Mixture subsets at 4 weeks, 12 weeks, and 20 weeks. Right column (B, D, F): Mixture subsets with spatial variability $\delta_r \in \{0.1°, 1.0°, 2.0°\}$. The color scale represents the ensemble standard deviation of a 6-year time-averaged occurrence per bin. The maps show the variability in particle dispersal patterns for all 50 subsets of the mixture simulations. The dashed line at $40°$W indicates the eastern boundary of the study area. The blue dot marks the approximate release location.



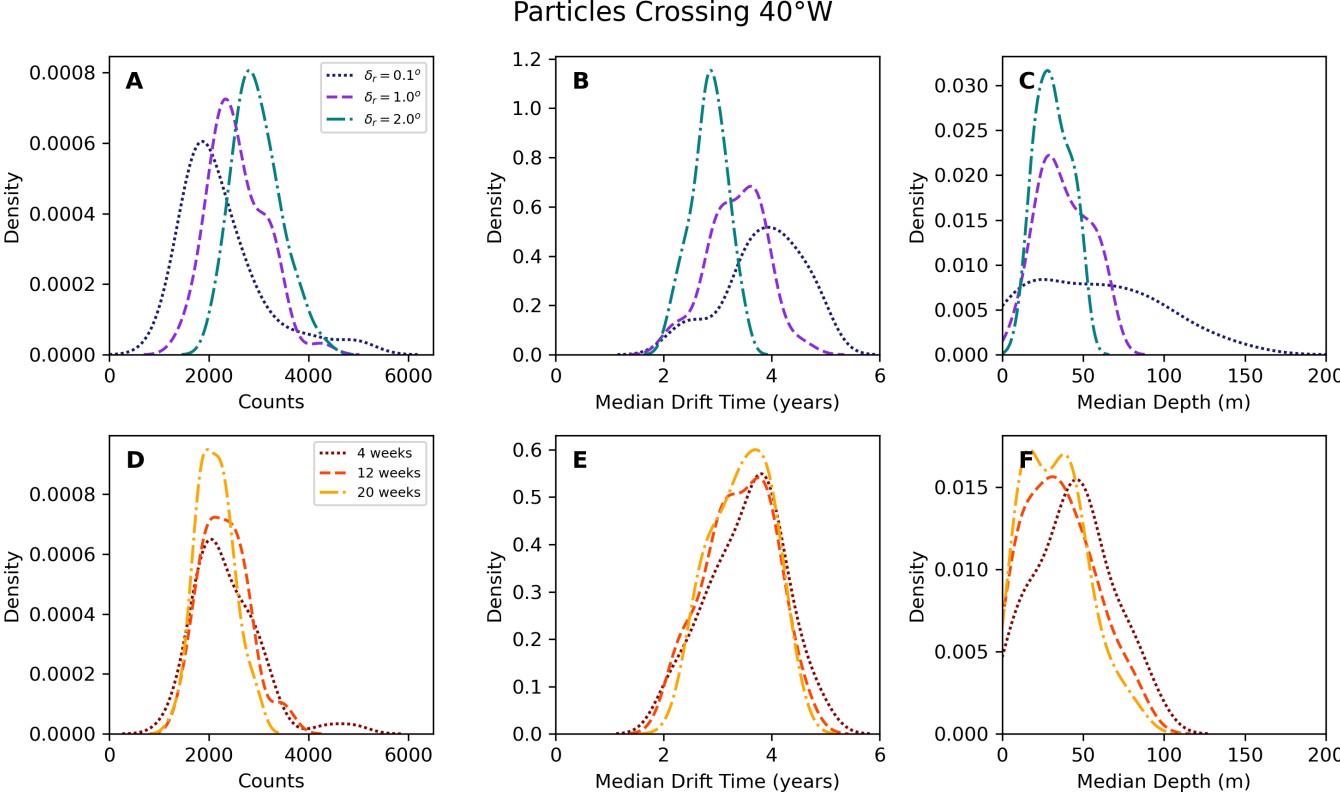

**Figure C3.** Kernel Density Estimates (KDE) of connectivity analysis for the single-member simulations. The top row (A-C) shows distributions for spatial releases $\delta_r \in \{0.1^\circ, 1.0^\circ, 2.0^\circ\}$: Particle counts (A), median drift time in years (B), and median depth in meters (C). The bottom row (D-F) shows the same metrics but is compared across different temporal releases of 4, 12, and 20 weeks.



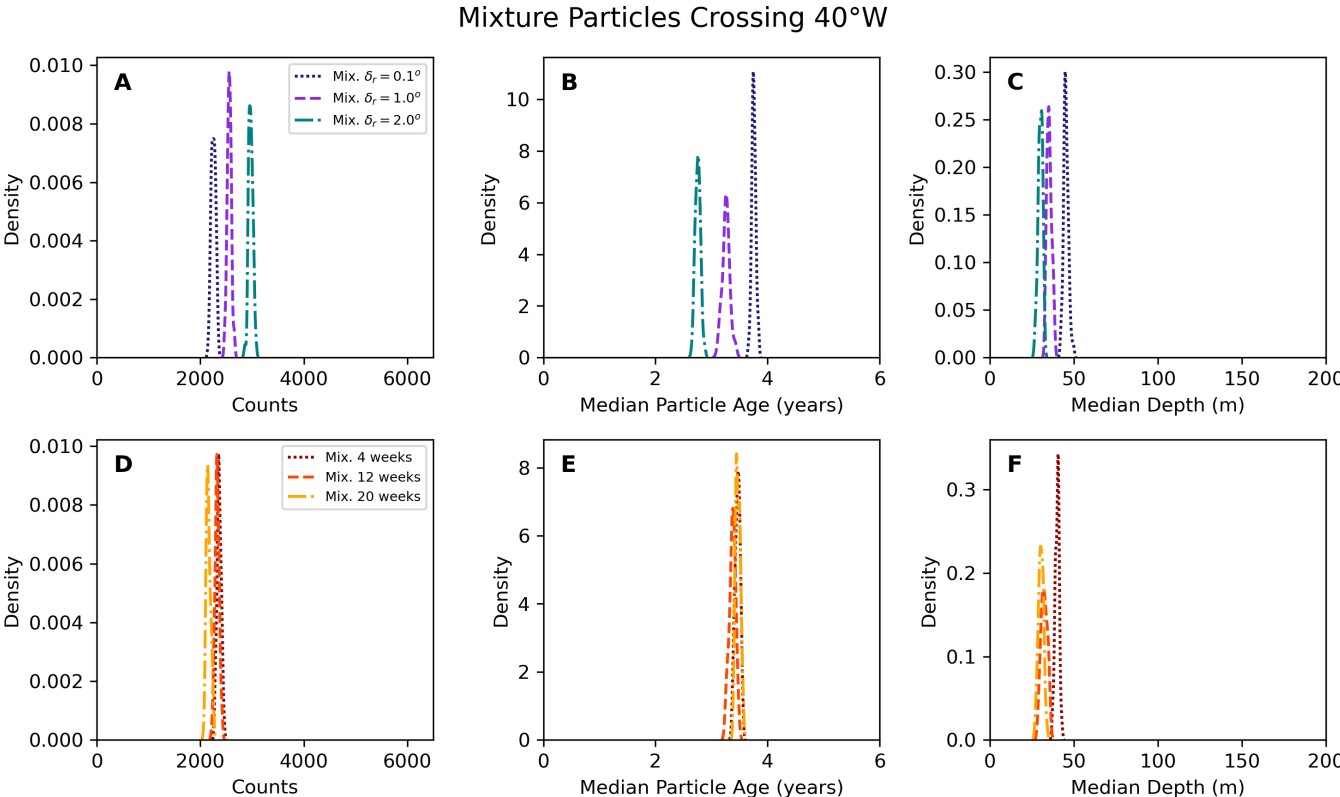

**Figure C4.** Kernel Density Estimates (KDE) of connectivity analysis for the mixture simulations, using Scott's method with a bandwidth of 1. The top row (A-C) shows distributions for mixture spatial releases $\delta_r \in \{0.1°, 1.0°, 2.0°\}$: Particle counts (A), median drift time in years (B), and median depth in meters (C). The bottom row (D-F) shows the same metrics but is compared across different mixture temporal releases of 4, 12, and 20 weeks.




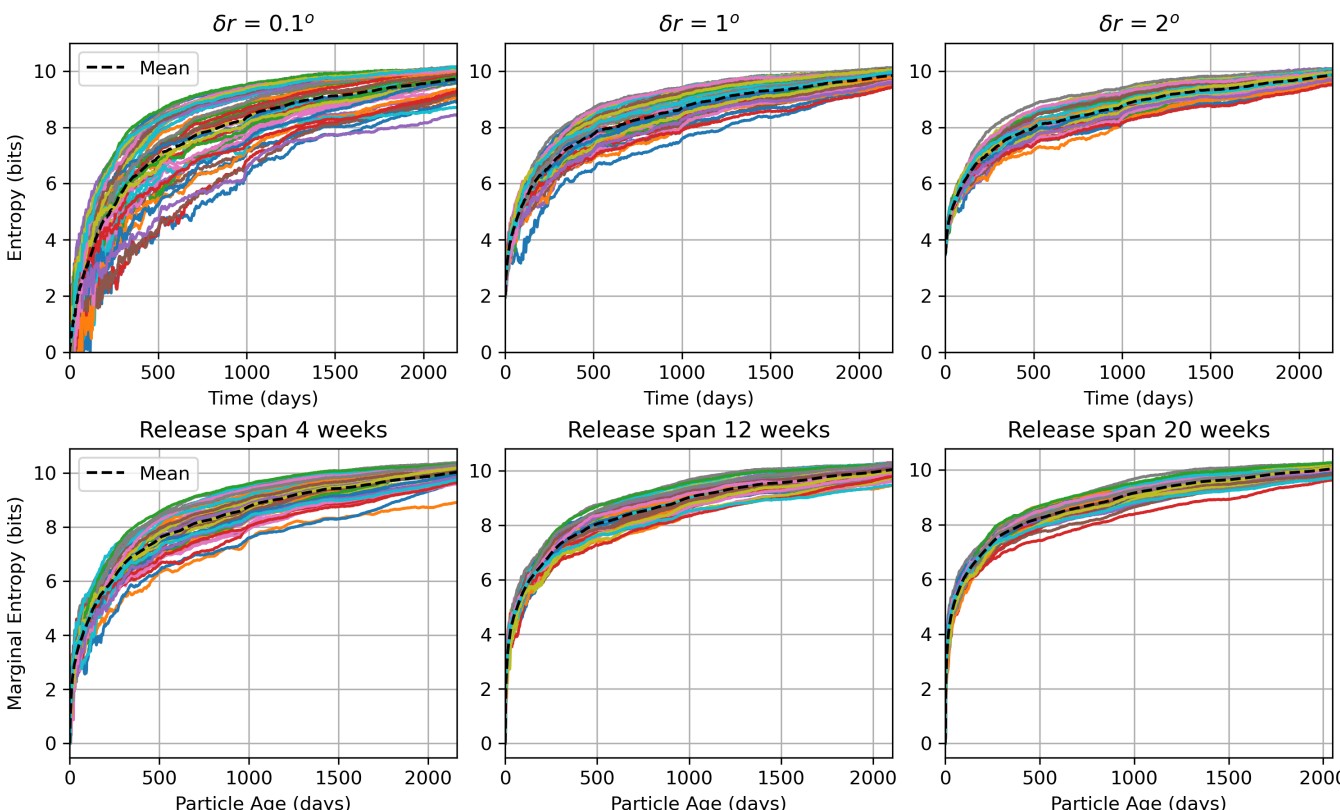

**Figure C5.** Marginal entropy the 50 single-member simulations performed with the different release strategies (individual panels). The lines are randomly-colored and each line represents one single-member entropy and the black dashed line is the ensemble average as a function of time.



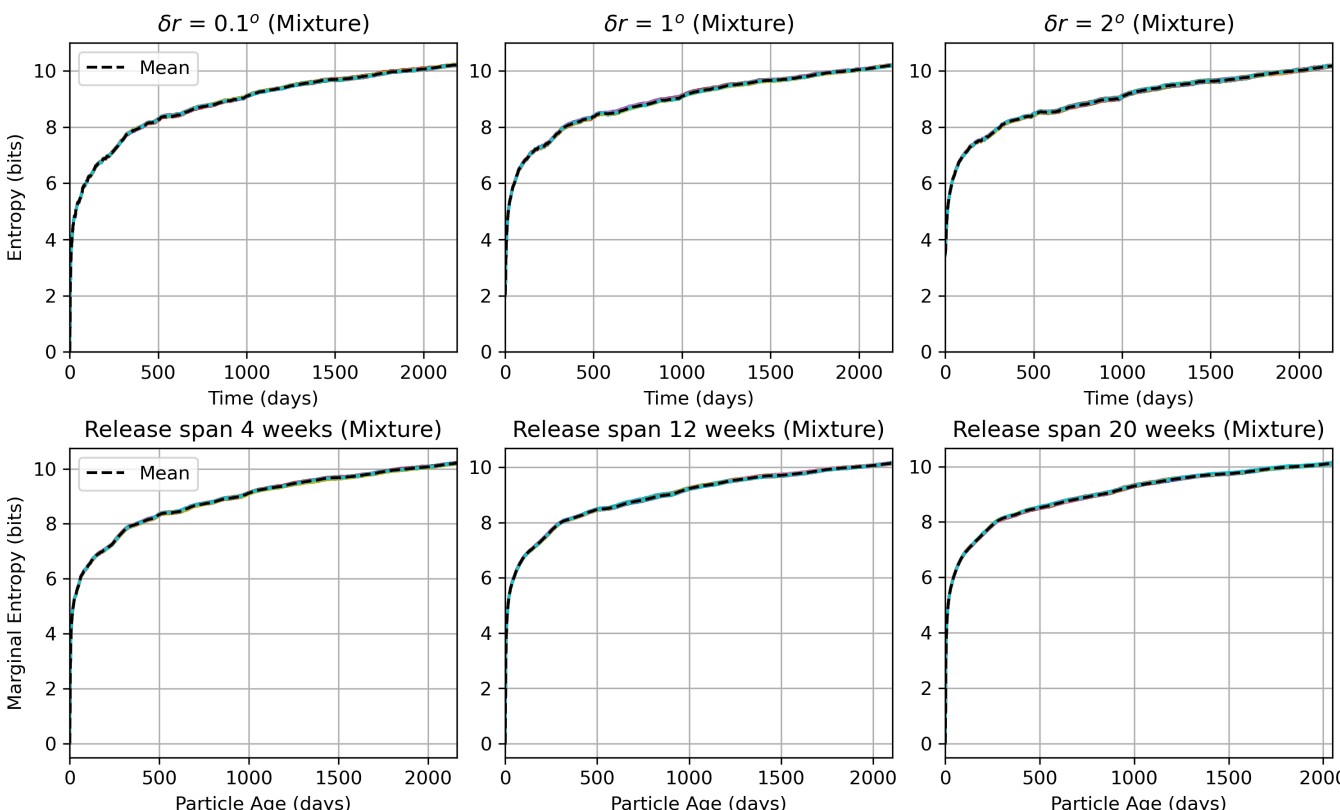

**Figure C6.** Marginal entropy of the 50 bootstrapping realizations or mixture distributions, subsampled from single-member simulations with different release strategies (individual panels). The color lines show the entropy of each realization, and the black dashed line is the average as a function of time.







**Figure C7.** Probability distributions of particle locations at different ages (10, 100, and 1,000 days; columns) across varying release strategies (rows). The top row shows the probability of the mixture $\delta_r = 0.1°$ distribution (subset 43 of the bootstrapping). The 2nd, 3rd, and 4th rows show the single-member distributions (member 22), with 20-week, $\delta_r = 2.0°$ and $\delta_r = 0.1°$ release, respectively. The blue circles mark the particle's release location, omitted for plots of particle age of 10 days. Bins with probability zero were removed to facilitate visualizing the area of dispersal of the particles.





**Figure C8.** Probability distributions of particle locations at different ages (10, 100, and 1,000 days; columns) across varying release strategies (rows). The top row shows the probability of the mixture $\delta_r = 0.1°$ distribution (subset 10 of the bootstrapping). The 2nd, 3rd, and 4th rows show the single-member distributions (member 46), with 20-week, $\delta_r = 2.0°$ and $\delta_r = 0.1°$ release, respectively. The blue circles mark the particle release location, omitted for plots of particle age of 10 days. Bins with probability zero were removed to facilitate visualizing the area of dispersal of the particles.



## Code and Data Availability

The data related to this article can be found at https://doi.org/zenodoXXXXX. The code is available at https://github.com/OceanParcels/NEMO_Ensemble_Lagrangian_Analysis.git.

## Author Contributions

CMP, SR, LGN, and EvS contributed to the conceptualization and design of this study. CMP performed the simulations and analysis and wrote the original draft. All authors contributed to the manuscript revision and read and approved the submitted version.

## Competing Interests

The authors declare that the research was conducted without any commercial or financial relationships that could potentially create a conflict of interest.

## Acknowledgments

CMP, FM, and EvS were supported by NWO through the grant OCENW.GROOT.2019.043. SR was funded by the European Union's Horizon Europe research and innovation program under the Marie Sklodowska-Curie grant agreement No 10110727. LGN acknowledges the funding of her Margarita Salas fellowship by the European UnionNextGenerationEU, Ministry of Universities and Recovery, Transformation and Resilience Plan, through a call from the University of the Balearic Islands (Palma, Spain). MCD was supported by the NECCTON project, which has received funding from Horizon Europe RIA (grant agreement No 101081273). This work is a contribution to the OCCIPUT project, which has been funded by ANR through contract ANR-13-BS06-0007-01. The ensemble simulation was performed using computer and storage resources provided by GENCI at TGCC thanks to the grant 2022-A0130112020 on the Joliot Curie supercomputer's KNL partition. The authors thank Jean-Marc Molines and Aurélie Albert for performing the ensemble simulation and distributing the model outputs.



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
