# Peer review of "Quantifying Variability in Lagrangian Particle Dispersal in Ocean Ensemble Simulations: an Information Theory Approach"

_EGUsphere, 2024_

## Author Comment (AC1)

Reviewer 1

This article works to obtain the variability in particle dispersion based on Eulerian velocity data, by generating variability in single (one initial condition) trajectories by choosing ensembles whose initial conditions are (i) close to the initial condition, and (ii) at same location as the initial condition, but are at nearby times. Information-theoretical measures are used to quantify the variability of these simulations in the Gulf Stream region.

Being able to gain insights into variability and uncertainty of simulations based on Eulerian velocity data is an important issue. To my mind, the crucial thing here would be in the ability to parameterize the input uncertainties (in the measurements of the Eulerian velocity, in using various forms of diffusivity in the advection process, in interpolation methods used to estimate data at subgrid scales where no data is available, etc), as well as give careful assessments and/or sensitivity analyses of the other variables (time-of-flow, radii used for initial seeding, release date, etc) in the process. Any quantifier of Lagrangian dispersal uncertainty obtained will of course be strongly dependent on the quantifiers of, and the perspectives on modeling the impact of, these input uncertainties. My main comments below are related to this main theme, on which any conclusions obtained from this study are crucially dependent.

We thank the reviewer for recognizing the important issue underlying our work. We have below carefully addressed all the comments.

1. The authors use spatial radii ranging from 0.1 to 2 degrees (9 to 180 km) for their initial clouds of particles in the spatial uncertainty situation. They quote that computed average decorrelation lengthscale of 0.41 degrees. It's not clear to me how long the particles were advected -- one would of course expect nearby fluid parcels to spread-apart more and more as time progresses (reflected in Figure 4A, for example, where the entropy is used rather than decorrelation). To my mind, these decorrelation numbers are therefore not "useful." What is the connection to this time-of-flow, and how does this decorrelation change in relation to that? Is the fact that 0.41 degrees comparable in size to the 0.25 degree of resolution in the NATL025-CJMCYC3 model, i..e., that the Eulerian velocity is not resolved below this lengthscale, relevant? What is the physical motivation for using distances of 9-180 km to seed particles in assessing the dispersal over a certain time for a single initial condition? Not highlighting (I don't know whether it's buried somewhere and I couldn't find it) the time-of-flow is a serious limitation, because this will have a profound effect on any results obtained.

These are all good questions, that we now clarify in the revised manuscript. Specifically:

1. We have now clarified in the main text that we computed the decorrelation length scale "*of the initial particle velocities at the release location on the release day (2 January 2010)*" (lines 144-146 of the track-changed pdf). We used this to inform the size of the spatial release patches because it gives an estimate of the flow scales *at the moment of release*, so we think it is a relevant metric.

2. We moved the description of the correlation time and length scales to the methods section to improve readability (as suggested by reviewer 2; lines 155-166 and lines 186-200 of the track-changed pdf). We moved Figure A1 to main text, as Figure 2. We also added the equation that we used for calculating the decorrelation length and time scales (Eq.(1) and (2) of the track changes document).

3. We clarified that these metrics were computed as particle-pair correlations to inform the correlation in the spatial and temporal varying initial conditions used to generate variability (line 152 and 186, respectively, of the track-changed pdf). However, we explicitly state, lines 169-172: "*While decorrelation scales likely evolve over the 6-year simulation period due to particle spreading and varying flow conditions, computing their evolution at every particle age is computationally intensive and impractical, compared to other metrics.*"

4. We have now clarified that the virtual particles were advected for six years (from 2 January 2010 to the end of the hydrodynamic data availability on 31 December 2015) – or slightly less in the case of the temporal varying release, where some particles were released up to 20 weeks later (lines 101 of the track-changed pdf). The time-of-flow is the same as the particle age that we use in e.g. Figures 3 and 4.

5. We also clarified in lines 139-141: "*The choice of spatial release radii (9-180 km) spans the range from sub-mesoscale to mesoscale oceanographic features, allowing us to test how initial condition uncertainties at different scales affect long-term particle dispersion.*"

2. It seems that all calculations were done with 2 January as the release date. Given the unsteadiness (time-dependence) of the Eulerian velocity field, one will of course get different results if using a different release date. From an oceanographer's viewpoint, why is 2 January so important? Alternatively, there needs to be convincing results to show that "similar" results are obtained for different choices of release dates. Also, one needs some understanding of the sensitivity of the results of the time-of-flow chosen.

The reviewer has a good point that there is nothing 'special' with 2 January 2010. We chose it for the practical reason that it allowed us the longest particle advection time given the length of the CJMCYC3 model fields. However, since our goal was to introduce an analysis methodology, we feel that a full analysis of how the results change for different release dates is beyond scope (and computational resources available).

We have now clarified this in the limitations paragraph of the discussion section of the revised manuscript: "*We performed simulations for only one release period, on 2 January 2010, because that allowed us the longest particle advection time; but realise that there is nothing 'special' about that 2 January 2010 and that the results presented here might depend on the release time*" (lines 653-656 of the track-changed pdf).

3.  For the temporally-varying release, the authors have used time windows of "4, 12 and 20 weeks, all starting from 2 January 2010."  Once again, the time-of-flow is not clear, and it is moreover not clear whether the time-of-flow for each of these releases is the same.  For example, if the time-of-flow was 2 weeks uniformly across all of these simulations, the "4 week time window" would, in my understanding correspond to having some particles released on 2 January and travelling  until 16 January, but then also particles released on 30 January and travelling until 13 February.  And presumably particles on intermediate days.  All of this has to be made absolutely clear to enable any assessment of the results obtained.  Moreover, as above, there needs to be some rationale for selecting these time windows.

Our advection strategy was apparently not clear in the first version of our manuscript. All particles travel until the end of 2015. We have now clarified in the methods section that "*all particles were advected until the end of 2015, so that in this simulation some particles reached a maximum 'age' (time-of-flow) of 6 years and others only 5.6 years. This is a minor effect though, as most of our analysis will focus on the first few months of advection only.*" (Lines 181-183 of the track-changed pdf).

4.  If my above interpretation of the particle release is correct, I think there are some issues which interfere with the interpretation of this process giving an indicator of Lagrangian dispersal.  Notice that in the example the particles travelling from 2 January to 16 January are driven by velocity field data which is *completely* different from the particles released on 30 January and travelling to 13 February. From the dynamical systems perspective, this means that one is comparing things which are driven by completely different dynamical systems -- so what does the usage of these different final locations in a dispersal assessment actually mean?  On the other hand, if the above interpretation is wrong, and there is some other way of thinking about this, what would the interpretation of that be?  For example, if all particles are flowed till 13 February despite being released on different days, then again the dynamical systems are different, as is the time-of-flow (which strongly impacts correlation and any measure of dispersal).  This sort of issue is relevant in the later calculations in the article as well, for example when computing time series of histograms described in Section 2.4---exactly how "time" is interpreted is crucial in an unsteady flow such as this.

As also mentioned in our previous response, we seem not to have been clear enough about our temporal release experiment strategy. The point of a temporally varying release is to seed particles that will have different trajectories, as the temporal decorrelation functions suggest. This is in fact the release strategy that is currently most widely used by Lagrangian ocean modellers to simulate different flow regimes, and the motivation of our study is to investigate how to assess whether this release strategy is equivalent to sampling different ensemble members. We have now further clarified this in the Introduction: "*Missing variability in particle trajectories is typically created by releasing particles at different locations (spatial variation; e.g., Rossi et al 2013) at different times (temporal variation; e.g., Qin et al 2014, van Sebille et al 2015) and/or with a small amount of random walk diffusion added to the advection (e.g., Hart-Davis and Backeberg 2021)*" (lines 64-70 of the track-changed pdf)

5.  The authors say "to ensure particles released on the same day followed different trajectories, we added small random perturbations to their release locations using uniform noise with an amplitude of 0.1 degrees."  Does this mean that one chooses uniformly from a ball of amplitude 0.1 degrees centered at the initial location?  What this actually means is that both temporal and spatial variability in the initial condition is used here, and not just temporal variability, does it not?

Yes, the reviewer is correct that indeed we added a small amount of spatial variability in the temporal-variability runs. However, as we also explain in that paragraph "*we kept this noise amplitude small to (as much as possible) isolate the effects of the temporal release strategy alone*" (lines 179-180 of the track-changed manuscript).

6.  Using temporal variability but from a set location is the concept of "streaklines" in fluid mechanics.  This has seen importance in assessing transport due to Lagrangian motion principally when a streakline can sensibly be used as some sort of "barrier" between coherent structures, notably when the position from which the streakline emanates is fixed (on a boundary), for example see Haller (J Fluid Mech 2004), Zhang (SIAM Review, 2013), Karrasch (SIAM J Applied Dyn Sys, 2016), Balasuriya (SIAM J Applied Dyn Sys, 2017).  For genuinely unsteady flows, since the velocity field around any fixed Eulerian location changes with time, the dispersal from a streakline approach (as the authors do here in their temporal release strategy) will provide a curve (a tube in this case since a small cloud is released at each time) at any given final instance in time -- but exactly how to interpret this tube from a Lagrangian dispersal perspective is unclear. It will consist of particles which have flowed for different times, and (for example) particles in a particular cross-section of the tube will not necessarily have flowed for the same time.  If instead the interpretation I have in point 3 above is used, what this means is that a particular subsample of the points in this "streaktube" will be obtained for each day of release -- and how one uses the collection of these subsamples to quantify Lagrangian dispersal appears ambiguous.

This is an interesting perspective by the reviewer. We have now added this relation to streaklines in the introduction section, where we also cite the papers suggested by the reviewer: "*In fluid mechanics, this is related to the concept of streaklines, transport barriers and coherent structures (e.g., Haller 2004, Zhang 2013, Karrasch 2016, Balasuriya 2017)*" (lines 51-52 of the track-changed manuscript).

7.  The authors do not seem to have used any randomness in the Lagrangian advection process (only randomness in initializing) which to my mind does not take into account  fundamental contributors to Lagrangian dispersal in the ocean: effects of eddy diffusivity and uncertainties in the driving velocity fields.  Eddy diffusivity modeling is a vast and important area (Berner et al, Bull Amer Meteorol Soc, 2017), for which many different models exist (e.g., Griffa, in Stochastic Modeling in Physical Oceanography, Birkhauser, 1996).  Here, though, dispersal seems to

happen though the taking of nearby initial conditions both spatially and temporally. This seems artificial when there is a "natural" way of including the primary physical issues via advecting a stochastic differential equation model with small noise, or using the alternative representation via the Fokker-Planck equation (e.g., Chandrasekhar, Rev Modern Phys, 1943) which explicitly governs a probability density of particles, or using more sophisticated eddy diffusivity models. The results from the Fokker-Planck, for example, would be an explicit quantifier of Lagrangian dispersal. There is a substantial literature on these methods, also in usage in oceanic data. Of course, running stochastic simulations of this nature, or attempting to solve the Fokker-Planck may incur substantial computational costs, but these would seem more compelling approaches to Lagrangian dispersal.

The reviewer has a good point here. In the revised manuscript, we have also added the results of a simulation where the particle spread was created by adding a small amount of diffusion on top of the advection. We described the method to add diffusion in section 2.3.3 in the revised manuscript, in lines 203-217 in the track-changed manuscript. As well, we updated Figures 4, 5, 6, 7 and 8 in the revised manuscript, to include the results from the added diffusion release strategy. As well, we updated the results section to include diffusivity (lines 397-399 and lines 514-519, of track-changed manuscript).

In the discussion section (lines 601-614 of the track-changed manuscript), we added: "*The added diffusion simulations use a stochastic differential equation approach with Brownian motion terms, following established methods in Lagrangian oceanography (Griffa, 1996). However, our results demonstrate that this approach has limitations in reproducing the full ensemble variability compared to spatial and temporal releases, particularly over particle ages < 100 days. We chose a physically reasonable diffusion coefficient based on literature values for subgrid-scale parameterization at our model resolution. Larger diffusivity coefficients could potentially increase trajectory variability, they would become unphysical as the artificial noise would exceed the magnitude of typical unresolved subgrid processes in the model, effectively forcing the particles to behave as if experiencing constant strong turbulent mixing. The underperformance of the diffusion strategy likely reflects that it represents primarily small-scale turbulent mixing processes, and the diffusivity cannot create noticeable variability in trajectories, specially for particle ages 100 days, missing the larger-scale flow uncertainties and mesoscale variability captured through other strategies.*"

8. There are emerging tools for avoiding stochastic simulation and explicitly obtaining some quantifiers which are related to dispersal, in particular the idea of stochastic sensitivity (Balasuriya, SIAM Review, 2020) which has been shown to also be robust when applying to oceanographic data (Badza, Physica D, 2023). The authors' approach here of seeking dispersal measures for each initial condition (as opposed to a large ensemble) at first glance appears to be similar to finding the stochastic sensitivity (a scaled variance of the final distribution around the deterministic final location, when the Lagrangian evolution is subject on ongoing model noise) at each member (initial) location. Another approach along these lines which appears relevant is that of Branicki and Uda (SIAM J Appl Dyn Sys, 2023).

The reviewer raises an interesting point here; these approaches are indeed nicely complementary. Hence, we have now added a few sentences to the introduction of our revised manuscript: "*Our approach is complementary to other new approaches for computing stochastic sensitivity of Lagrangian trajectories in the ocean, such as those by Balasuriya (2020), Badza et al. (2023) and Branicki and Uda (2023). However, our approach is particularly also useful for particles with added `behaviour', such as in the case of plastic particles (e.g., Denes and Van Sebille, 2024)*" (lines 73-76 of the track-changed pdf).

9. The authors use several quantifiers for dispersal based on their Lagrangian simulations: mixture probability distributions, connectivity, entropy, and KL-divergence. Computing each of these requires the final Lagrangian distributions. So, interpreting any of these depends strongly on how those Lagrangian distributions were obtained in a physically reasonable fashion (see my earlier points on this). Because of this, it is hard for me to interpret any of these information-theoretic results.

We agree with the reviewer that the fidelity of our results depends crucially on the details of the Lagrangian integration scheme. We expect that the changes we made in response to other comments already help to clarify our methodology, and to even further highlight where details on the trajectory computations can be found, we have restructured the Methods section so that there is now a separate section (2.2) "*Lagrangian simulations*" (line 101 of the track-changed pdf).

10. It appears that a major point that is being made is that the full-ensemble variability is being approximated by single-member simulations. This is stated in many places, but I am a little uncertain as to whether my interpretation of this statement is correct. My understanding is that "single-member" means one particular initial condition is chosen. By choosing "50 members" (line 158), Lagrangian simulations associated with 50 different initial conditions are chosen. Then the probability distributions associated with each of these 50 are combined in a mixture model to get the "full" distribution. Is this understanding correct? If so, it would appear that the "full ensemble" can be thought of as the 50 simulations collectively, and so this is collectively the "full ensemble"? If this is so, the claim that the "full-ensemble" is obtained from "single-member" simulations, presumably demonstrating some advantage, isn't too different from simply saying that one is choosing a full-ensemble comprising 50 members, presumably chosen to cover a region of interest at the initial time sufficiently well? If this is not true, things have not been expressed clearly. Basically, I think some clarification on this claim is necessary, explaining exactly what is meant and how it is achieved (and exactly what a "single-member simulation" means).

The reviewer is right that the term full-ensemble variability is somewhat unclear. In the revised manuscript, we have replaced it with "*variability of the full 50-member ensemble*" (lines 6 in the abstract) and "*Our goal is to reproduce this dispersion of what we refer to as the `full ensemble' using just one ensemble member*" (lines 115 of the track-changed manuscript)

11. And finally to return to my preface to the numbered points: to interpret any of the results on dispersal, one needs to be comfortable that the strategies for computationally determining dispersal here have something to do with the physical issues which lead to dispersal. Many of the points above are related to this, asking for clarification as to why the actions used in this article to generate dispersed trajectories are meaningful, whether they can be parameterized in terms of something physical, effects of time, why diffusivity/stochasticity in the evolution is ignored, why other techniques which explicitly capture dispersion are not used, etc.

We agree with the reviewer that the fidelity of the results depend on the quality of the Lagrangian integration scheme and choice to include pure advection as well as stochastic approaches. We hope that our responses above have convinced the reviewer that this is the case. However, we would also like to highlight that our proposed analysis approach would have value by itself to other numerical oceanographers wanting to analyse their own ensemble simulations, for their own region of interest.

Based on my comments above, I feel that a major revision would be necessary for this article to be acceptable for publication in Nonlinear Processes in Geophysics.

We hope that our changes help convince the reviewer of the quality of our work, and thank them again for the very useful comments and ideas that have significantly improved our manuscript.

---

## Author Comment (AC2)

1. The manuscript concerns ensembles of trajectories of tracers generated by ocean models. The authors wish to generate the "variability" of multiple ensembles by manipulating a single trajectory. The manipulations are primarily perturbing an initial condition in space, or by starting the trajectory at a different time. In order for such a study to be useful, pinning down exactly what "variability" means is crucial. The word "variability" is used extensively in the first several pages of the manuscript without stating what it actually is;  I think perhaps on page 7, in relation to "connectivity", the reader begins to see what might be meant. As far as I could tell, according to this manuscript, variability means either "connectivity", or various types of entropy of coarse-grained future distributions of trajectories. A case is not clearly made for why these quantities are useful for ocean dynamicists or oceanographers. That is, why are these quantities the gold standard by which oceanographers should assess "sameness" of (collections of) trajectories.

We thank the reviewer for this important comment; they are absolutely right that the term "variability" needs a clear definition. In the revised manuscript, we have added the paragraphs below (lines 34-47 in the track-changed manuscript):

"*However, similar to above, the trajectories obtained from the particle tracking in a single OGCM ensemble member may not be representative of the full probability density of the system's state. Because pure advection is deterministic, there will be only one trajectory resulting from a virtual particle that starts at a certain place and time.*

*This deterministic nature limits what we define as 'trajectory variability' – the range of possible pathways and end locations that particles could follow given uncertainties in ocean conditions. We define trajectory variability as the spread in particle positions, pathways, and connectivity patterns that emerges when accounting for uncertainties in initial conditions or modelled ocean states.*

*Capturing the trajectory variability is crucial for practical oceanography applications. For example, search and rescue professionals may want to compute a full probability density function of possible object locations – even when the starting location and time of an object lost at sea is known exactly – due to uncertainties in the ocean model. Similarly, marine pollution studies need to assess the range of possible contamination pathways, while connectivity studies in marine ecology require understanding the full spectrum of larval dispersal routes between habitats. In each case, a single deterministic trajectory provides insufficient information, limiting the generalisability of the results, as it cannot represent the inherent uncertainty in ocean dynamics and model predictions.*"

2. The manuscript does not mention models where subgrid-scale dynamics is simulated e.g. stochastically. This would appear to be a very relevant set of comparators. It is also well known that trajectories are influenced by the resolution of the model grid, and that very different dynamics

can arise from the same model with different resolutions. This aspect is also not addressed; as far as I understand, only a single 1/4 degree model is used.

This point was also mentioned by reviewer 1, and we have now added a third set of experiments to the manuscript where we simulate particles with some Brownian diffusion added (lines 203-217 in the track-changed manuscript). We as well updated the results and discussion section to include these results (lines 397-399 and lines 514-519, respectively, in the track-changed manuscript)

Line 90: "The first strategy varies the release locations". Isn't this exactly part of what one does with ensemble generation? What is the difference?

We think there may be some confusion here between model trajectories in ensemble modelling and particle trajectories in Lagrangian analysis. We have thus now changed this to "*varies the release locations of the virtual particles spatially*" (lines 117-118 in the track-changed manuscript).

Section 2.2: I believe it is a poor choice to put the first results in an appendix. In fact, the manuscript reads as though it was recently chopped up and rearranged because it is impossible to read from start to finish via the Appendices. The Appendices contain definitions and details that the reader has not yet come across when reading from start to finish. I would strongly recommend removing the appendices and putting the material in the body of the paper to help with the narrative flow.

We have now moved the definition of the decorrelation time and length scales to the methods section of the manuscript (lines 152-166 and lines 186-200 of the track-changed manuscript). We also moved the Figure B5 (Figure 5 in the track-changed and updated manuscript) out of the appendix, showing the probability distributions of one ensemble member at different ages with different strategies, to the results section and adding a description of the figure in lines 428-443. This figure helps illustrate the probability distributions and the subsequent quantitative analyses for comparing the distributions. These modifications improved the readability of the manuscript.

Section 2.4: I could not understand how the probability distributions were being formed. The description is wordy, vague, and a bit sloppy. It needs precision and some formulas wouldn't hurt. Is a hexagon a bin? I could not find it stated.

We appreciate the observation. Based on it, we rewrote the sections describing the probability distributions and mixture probability distributions, by adding equations describing how we computed this distributions (Eq.(4) and Eq.(5) and lines 219-275, of the track-changed manuscript). We also added that these bins are indeed the same as the hexagons, by referring specifically to "*(hexagonal) bins*" (e.g lines 299 and 355 in the track-changed manuscript).

Section 2.5: I could not understand what a mixture probability distribution is. This seems to be a crucial object in the manuscript, but the description was brief and ambiguous. Again, some formulas may help. There is a discussion about the optimal number of particles. In what sense optimal? Again the reader is referred to an Appendix that is too brief and does not provide any insight.

As stated previously, we have added some extra mathematical formulations to better explain the concept of the mixture probability distribution (lines 247-275 of the track-changed manuscript). We also made sure to stay consistent with the mathematical notation throughout the manuscript to avoid confusion.

Section 2.6: Connectivity is not defined, and it is not explained in the Appendix. What is it?

We have now clarified in the Introduction section that we define connectivity as "*a metric that maps the origin of substances (water, nutrients, plankton, plastic objects) to their destinations*" (lines 30-31 of the track-changed pdf).

Section 2.7: Similar to section 2.4, the section is written verbosely and somewhat sloppily, to the extent that I could not understand what the various definitions were.

We improved the section 2.7 "*Marginal Entropy and Relative Entropy Calculation*" (section 2.4.4 in the track-changed manuscript, lines 296-371) by homogenizing mathematical notation and variables used in the previous sections describing the probability distributions, to stay consistent and avoid confusion. This removes some ambiguities in the explanations of the concepts marginal and relative entropy. In general, we followed the notation and explanation style of Shannon's 1958 article, which is a is a mix of verbose and mathematical expressions.

Line 202: "ensemble of bins" what does this mean?

We now realise that using the word "ensemble" for anything other than the NATL025-CJMCYC3 ensemble could lead to confusion. We have thus replaced these uses of ensemble by "*set*" (e.g. lines 141, 152, 309 of the track-changed pdf).

Line 203: "t is the particle age of the distribution". This may make sense if the distribution was unambiguously defined at some point.

In the revised manuscript, we now mathematically define the probability distribution (lines 241 of the track-changed pdf).

Line 210: The authors write "P_A(X) = (1/2, 1/4, 1/8, 1/8)".  As far as I understood from e.g. line 201, P(x_i) should be the probability of event x_i occuring, i.e. a number. Therefore since X=(x_1,x_2,x_3,x_4) -- see line 211 -- P_A(X) should equal 1, not a string of probabilities.  This is just one example of the vague writing. If the authors really want P_A(X) to be a string of probabilities, that is fine, define some suitable object, and ensure that the writing is clear and consistent.

We thank the reviewer for highlighting this issue. We stated that the set of hexagonal bins is defined as $X = (x_1, .., x_B)$, where $x_i$ is a hexagonal bin, and $B$ is the number of bins in the domain (see lines 229-230 of the track changes pdf). Following this notation $P(X)$ represents the probability distribution over all hexagonal bins and $P(x_i)$ is the probability of finding particles in one bin. This was added in lines 243-245 of the track changed manuscript.

In summary, on the basis of both the writing and the scientific impact, I do not recommend publication.

We thank the reviewer for their comments, and hope that our responses have clarified some confusion so that the reviewer can be more supportive of this revised version.

The manuscript could be improved by going back to the drawing board and asking what exactly are the properties by which it is *most meaningful* to oceanographers to compare trajectories or ensembles of trajectories.  Strong justifications and illustrations would need to be provided. Comparing with ocean models of different types, including stochastic components, and across different grid resolutions would add to the robustness and generalization of the subsequent results.  A linear narrative and a much more precise presentation would also be required.

This is a good point by the reviewer, that we have taken on board. We think that in this revised manuscript, we have balanced further clarification of what many oceanographers find meaningful (connectivity!) and that the added comparison to adding Brownian motion is meaningful.

---

## Author Response (AR2)

**Reviewer 3**

This study uses numerical simulations to compare Lagrangian statistics in two configurations: (i) an ensemble of simulations in different realizations of the flow, and (ii) simulations in the same flow realization, but with different release times and locations. In the first case, the flow realizations can be assumed independent. In the second, the flow remains correlated in time and/or space across the different releases. It is my understanding that the central question is how much these correlations influence the overall statistics of Lagrangian trajectories. The second configuration is particularly relevant to practical applications, since in the real ocean, drifter releases may occur at different times and locations but within the same flow realization. This is an interesting and important topic, and the study provides a thorough and valuable analysis.

I have a few comments and requests for clarification listed below, but overall, I recommend publication following revision.

We thank the reviewer for this very encouraging view on our study. We have below carefully addressed all the comments.

**I.93: Please briefly clarify how this perturbation is applied.**

We have now expanded the explanation of how the OCCIPUT developers applied this perturbation: "The inter-member dispersion was generated by activating small stochastic perturbations in the density gradients resolved by the model during 1993, and deactivating these perturbations for the remaining simulation time, as presented in Bessières et al (2017) and based on the algorithm of Brankart et al. (2013)" (lines 95-98 of the track-changed pdf).

II.135–136: The model has an eddy-permitting resolution of ¼°, and the shortest meaningfully simulated spatial scale is most likely longer than 100 km, which implies that submesoscale motions are not represented. Please clarify why the release radius was made as small as 9 km, and what is meant by "submesoscale features" in this context. It is not surprising that particles released within 9 km and even 90 km radii exhibit only modest dispersion. I suggest the authors include a discussion of the spatial scales resolved by the model and how they relate to the choice of small release radii.

We thank the reviewer for this good comment. Indeed, our goal was to test the sensitivity in relation to the grid scale. We have now rephrased this sentence in the revised manuscript to "The choice of spatial release radii (9 km-180 km) spans the range from sub-grid scales to ten grid cells apart, allowing us to test how initial condition uncertainties at different (grid) scales affect long-term particle dispersion" (line 139 of the track-changes pdf).

I.160: This comment echoes my earlier concern. At this resolution, the shortest resolved spatial scale exceeds 100 km, so the meaningfulness of the estimates at smaller spatial scales is unclear.

We agree that the way we linked the numerical simulation to (sub)mesoscale ocean dynamics in the real ocean may have been confusing. We have therefore removed the reference to the Rossby radius of deformation in the revised manuscript (lines 162-163 of the track-changed pdf).

I.199: The choice of Kh and the purpose of this experiment require further justification. While such a low diffusivity may indeed be used in simulations at 25 km resolution, here it appears the diffusion is intended to mimic ensemble variability due to resolved mesoscale motions, rather than unresolved subgrid-scale processes. As such, the very limited spread of trajectories in Fig. 1d is not surprising. A much larger value, such as 1,000 m²/s or more—as used in coarse-resolution simulations—would likely be more appropriate. I would recommend a simulation with high diffusivity.

This is a very good suggestion by the reviewer. In the revised manuscript, we have included another set of simulations for  $K_h = 1000 \text{ m}^2/\text{s}$ , and compare them with the  $K_h = 10 \text{ m}^2/\text{s}$  simulations throughout the manuscript. Notably, the simulation with  $K_h = 1000 \text{ m}^2/\text{s}$  performs remarkably well, with respect to the entropy but not at all with respect to the connectivity. We now also highlight that in the abstract and conclusions (lines 14-16 and 584-596 of the track-changed pdf).

I.235: Is "mixture distribution" the same as the probability distribution referenced in Equation (5)? This is somewhat unclear. Since the mixture simulation is intended to represent the full ensemble, while this section discusses a single-member simulation, clarification would be helpful.

The reviewer is right that the relation between these two mathematical concepts should have been clearer. In the revised manuscript, we now extended the sentence to "We then used these probability distributions to compute the mixture distributions over all grid cells  $\mathbf{P}_{\mathrm{mix}}(X|r,t)$  for each single-member strategy ..." (lines 244-245 of the track-changed pdf).

I.340: This sentence needs clarification. While I agree that choosing a release radius smaller than the model's spatial resolution effectively results in a point release, it is unclear what is meant by "full ensemble variability" in this context.

We have now clarified this sentence to "Throughout this analysis, we use the mixture distribution with  $\delta_r = 0.1^{\circ}$  as our reference, as it represents the closest approximation to a point release, while still being controlled by the variability in ocean velocities from the full ensemble variability" (lines 350-351 of the track-changed pdf).

I.460: It would be helpful to remind the reader how to interpret the results shown in Fig. 7, and to explain why low values of relative entropy indicate a good approximation of the reference case.

We have now added a sentence to the revised manuscript to help the reader interpret the results of Fig 7: "In these plots, low values of the relative entropy indicate a good agreement to the reference case, as relative entropy is a measure of the mismatch between each of the distributions shown and the reference distribution (technically: the cost of assuming that each of the distributions shown is the reference distribution)" (lines 493-495 in the track-changed pdf).

I.555: While I agree with the conclusion stated here, the purpose of this experiment remains unclear. If the goal is to mimic mesoscale-induced ensemble spread, it would make more sense to use a much larger diffusivity.

With the inclusion of the new simulations with  $K_h = 1000 \text{ m}^2/\text{s}$ , we can now much better reflect on the underperformance of the  $K_h = 10 \text{ m}^2/\text{s}$  simulations, so we thank the reviewer again for this suggestion.

I.574: Please remove the extra "the."

We have removed the repeated "the" from the revised manuscript.

---

## Author Response (AR3)

**Reviewer 3**

The authors adequately addressed my previous suggestions and comments, and, in my opinion, the paper is ready for publication.

An additional suggestion is to briefly present the rationale for using a high diffusivity value of Kh = 1000 m²/s. One could argue that, since the ensemble spread is largely driven by partially resolved mesoscale variability in the model, the chosen diffusivity should be sufficiently high to reflect eddy-induced particle dispersion—an effect that can be estimated through Lagrangian analysis.

We thank the reviewer for this final comment. In the revised manuscript, we have now added a sentence in the description of our diffusion experiments to explain the rationale for Kh=1000 m2/s: "The second is a high diffusion of  $K_h = 1000 \text{ m}^2 \text{ s}^{-1}$ , which is a value used to parameterize eddies in ocean models with  $O(1^\circ)$  spatial resolution (Reinders et al 2022). This latter value likely overestimates the eddy-induced particle dispersion driven by the partially resolved mesoscale variability in the ensemble members, so can be considered an extreme case." (lines 202-205 of the track-changed pdf).